# Time-series of Landsat-based bi-monthly and annual spectral indices for continental Europe for 2000–2022

Xuemeng Tian[1,2], Davide Consoli[1], Martijn Witjes[1], Florian Schneider[3], Leandro Parente[1], Murat Şahin[1], Yufeng Ho[1], Robert Minařík[1], and Tomislav Hengl[1]

[1]OpenGeoHub, Doorwerth, The Netherlands
[2]Laboratory of Geo-Information Science and Remote Sensing, Wageningen University & Research, Wageningen, The Netherlands
[3]Thünen Institute of Climate-Smart Agriculture, Germany

**Correspondence:** Xuemeng Tian (xuemeng.tian@opengeohub.org)

**Abstract.** The production and evaluation of the Analysis Ready and Cloud Optimized (ARCO) data cube for continental Europe (including Ukraine, the UK, and Turkey), derived from the Landsat Analysis Ready Data version 2 (ARD V2) produced by Global Land Analysis and Discovery team (GLAD) and covering the period from 2000 to 2022 is described. The data cube consists of 17TB of data at a 30–meter resolution and includes bimonthly, annual, and long-term spectral indices on various thematic topics, including: surface reflectance bands, Normalized Difference Vegetation Index (NDVI), Soil Adjusted Vegetation Index (SAVI), Fraction of Absorbed Photosynthetically Active Radiation (FAPAR), Normalized Difference Snow Index (NDSI), Normalized Difference Water Index (NDWI), Normalized Difference Tillage Index (NDTI), minimum Normalized Difference Tillage Index (minNDTI), Bare Soil Fraction (BSF), Number of Seasons (NOS), and Crop Duration Ratio (CDR). The data cube was developed with the intention to provide a comprehensive feature space for environmental modeling and mapping. The quality of the produced time series was assessed by: (1) assessing the accuracy of gap-filled bimonthly Landsat data with artificially created gaps, (2) visual examination for artifacts and inconsistencies, (3) plausibility checks with ground survey data, and (4) predictive modeling tests, examples with soil organic carbon (SOC) and land cover (LC) classification. The time series reconstruction demonstrates high accuracy, with RMSE smaller than 0.05, and $R^2$ higher than 0.6, across all bands. The visual examination indicates that the product is complete and consistent, except for winter periods in northern latitudes and high-altitude areas where high cloud and snow density introduce significant gaps, and hence many artifacts remain. The plausibility check further shows that the indices logically and statistically capture the processes. The BSF index showed a strong negative correlation (-0.73) with crop coverage data, while the minNDTI index had a moderate positive correlation (0.57) with the Eurostat tillage practices survey data. The detailed temporal resolution and long-term characteristics provided by different tiers of predictors in this data cube proved to be important for both soil organic carbon regression and LC classification experiments based on the 60,723 LUCAS observations: long-term characteristics (tier 4) were particularly valuable for predictive mapping of SOC and LC coming on the top of variable importance assessment. Crop-specific indices (NOS and CDR) provided limited value for the tested applications, possibly due to noise or insufficient quantification methods. The data cube is made available under a CC-BY license and will be continuously updated.

# 1 Introduction

Earth Observation (EO) data is increasingly recognized for its critical role in environmental sciences (Kansakar and Hossain, 2016; Salcedo-Sanz et al., 2020). Compared to traditional field surveys, it offers high-resolution, large-scale, and recurrent spatial environmental data at relatively low cost. These capabilities are important for comprehensive environmental studies, ongoing monitoring, and effective management (Chatenoux et al., 2021; Giuliani et al., 2021). EO data not only offer valuable insights into the Earth's surface through various spectral bands but can also highlight specific land surface features through spectral indices derived by processing them. In addition, satellite data serve as an essential input for various models that study physical processes, forecast future scenarios, and inform policy- and decision makers (Salcedo-Sanz et al., 2020).

As EO technology develops, more opportunities emerge. The development of a wider range of spectral indices allows a more detailed analysis using different combinations of satellite bands (Montero et al., 2023). Improvements in spatial and temporal resolutions enable more detailed observations. In addition, a longer record of satellite imagery facilitates long-term environmental studies. These advances help us better understand the environmental dynamics and enhance natural resource management strategies. However, they also introduce new challenges. One of the key challenges to fully unlock the potential of EO data for environmental applications, as highlighted by numerous researchers (Killough, 2018; Wagemann et al., 2021; Giuliani et al., 2021; Chatenoux et al., 2021; Montero et al., 2023), is the gap between the demand for detailed EO data and the limited processing capabilities of most users. As the volume of EO data and the complexity of spectral indices increase, processing typically requires specialized expertise and costly computational resources, both locally and in the cloud. For example, while platforms such as MODIS, Landsat, and Sentinel openly provide valuable satellite data, these data often require preprocessing to remove poor quality pixels, such as those affected by cloud cover and geometric distortions (Wulder et al., 2022; Radeloff et al., 2024). While many pre-processed layers are available, they tend to be scattered across various data portals and often focus on limited themes. Accessing these ready-to-use datasets usually demands domain-specific knowledge to even be aware of their existence. In addition, non-standardized data formats further complicate the ability of users to integrate these data sets for specific applications (Lokers et al., 2016; Wagemann et al., 2021). Thus, there is an essential need for solutions that ensure easy access to open environmental data. Furthermore, these solutions should also facilitate the easy integration of these data, enabling its combined use to effectively address critical environmental and economic challenges (Giuliani et al., 2017).

Numerous platforms have emerged in the last decade that aim to make the management, processing and analysis of big EO data operational, including Google Earth Engine (GEE, Gorelick et al., 2017), Sentinel Hub (SH, https://www.sentinel-hub.com/), Open Data Cube (ODC, https://www.opendatacube.org/), Copernicus Data Space Ecosystem (https://dataspace.copernicus.eu/), and openEO Cloud (https://openeo.cloud/). Each platform has its strengths and weaknesses: GEE offers extensive datasets and robust processing, but is limited by its closed nature, restricting external contributions and governance. Sentinel Hub faces

similar limitations. OpenEO provides flexibility with user-defined functions but lacks guaranteed compatibility across different back-end systems. Among these, the Open Data Cube (ODC) stands out for its open-source nature, flexibility, and strong community support, making it a leading option (Gomes et al., 2020). As a result, many analysis-ready data (ARD) cubes have been developed using ODC principles. ARD refers to a multidimensional time series stack of spatially aligned pixels that are ready for analysis, eliminating the need for additional data harmonization (Giuliani et al., 2017, 2021). These efforts reduce technical complexity and have been shown to be effective in delivering information efficiently, as demonstrated by successful implementations in various domains, such as vegetation dynamics (Obuchowicz et al., 2023), snow cover (Poussin et al., 2023), and drying trend (Poussin et al., 2021).

EcoDataCube.eu, developed by Witjes et al. (2023), is the first product to cover the entire EU with a sufficient temporal range to support a long-term analysis of land degradation and land use change dynamics. This work extends the concept of ARD to ARCO (analysis-ready and cloud-optimized) data cubes (Stern et al., 2022; Iosifescu Enescu et al., 2021), which helps to optimize cloud-based data management and processing. Cloud optimization allows for on-the-fly access, reduces latency through partial and parallel reads, and efficient metadata handling. Cloud-Optimized GeoTIFF (COG, https://www.cogeo.org/), adopted by EcoDataCube.eu, is a good example of such a format. COG files are structured to facilitate network access through HTTP range requests, ensuring compatibility with cloud object storage systems. This design supports integration with high-level analysis libraries and distributed frameworks. Additionally, EcoDataCube.eu also minimizes data gaps caused by clouds, which is a major obstacle to unlocking the potential of EO data (Baumann, 2024). This is achieved through time-reconstruction algorithms to ensure optimal cloud-free conditions (Zhou et al., 2016; Consoli et al., 2024). This approach not only prepares the data for spatio-temporal analysis but also proves beneficial for modeling, as models typically require complete input data to function effectively.

EcoDataCube.eu provides valuable insights through its quarterly temporal resolution and raw bands, making it a significant resource for Earth observation applications. To further reduce the burden on end users and increase the use and impact of EO data, we developed the ARCO data cube for the EU by refining the temporal resolution, processing reflectance bands to derive spectral indices, and updating the data layers until 2022. Our Landsat-based spectral indices data cube spans from 2000 to 2022, covering the pan-European area, including Ukraine, the UK, and Turkey, with data at a detailed 30m resolution. Using the Global Land Analysis and Discovery team (GLAD) Landsat Analysis Ready Dataset version 2 (ARD V2) presented by Potapov et al. (2020) as input, we adopt the state-of-the-art approach of Consoli et al. (2024) to generate four tiers of environmental predictors:

1. Bimonthly aggregated cloud-optimized bands;

2. Bimonthly spectral indices derived from bands;

3. Annual indices derived by analyzing the bimonthly time series of indices;

4. Long-term indices reflecting features across two decades.

The foundational tier of the data cube consists of bimonthly cloud-optimized bands, derived using time reconstruction methods to address the gaps left by cloudy pixels (Consoli et al., 2024). Building upon this, we calculate bimonthly spectral indices as the second tier of predictors. This selection includes the most widely used indices, covering key aspects such as vegetation, crops, soil, and water (Montero et al., 2023): the Normalized Difference Vegetation Index (NDVI) is used to evaluate vegetation health and biomass. Complementing NDVI, the Soil Adjusted Vegetation Index (SAVI) has been shown to be more effective in areas with sparse vegetation by minimizing the impact of soil brightness on vegetation sensing (Huete, 1988; Rhyma et al., 2020; Reddy et al., 2022). Additionally, the Fraction of Absorbed Photosynthetically Active Radiation (FAPAR) provides a more direct measurement of plant productivity (Myneni and Williams, 1994). The Normalized Difference Water Index (NDWI), also known as the Normalized Difference Moisture Index (NDMI), provides insights into water dynamics and climatic characteristics (Gao, 1996). The Normalized Difference Snow Index (NDSI) helps identify snowy areas (Salomonson and Appel, 2006). The Normalized Difference Tillage Index (NDTI), also known as Normalized Burn Ratio 2 (NBR2), shows potential in tillage detection, post-fire recovery studies and soil sealing identification (Zheng et al., 2012; Daughtry et al., 2010; Eskandari et al., 2016; Beeson et al., 2020; Sonmez and Slater, 2016; Storey et al., 2016; Ettehadi Osgouei et al., 2019; Xiang et al., 2022).

Through time series analysis, we also derive several annual indices that capture temporal patterns within the year as our third tier of predictors. This includes the Number of Seasons (NOS) and Crop Duration Ratio (CDR) from NDVI time series, which shed light on the intensity of cropland use (Siebert et al., 2010; Li et al., 2014; Patel and Oza, 2014; Estel et al., 2016). The Bare Soil Fraction (BSF) measures the duration of soil exposure, which is related to soil health (Demattê et al., 2020; Mzid et al., 2021; Sharma et al., 2018; Turmel et al., 2015). Although NDTI is positively correlated with the crop residue cover ratio (CRC), using it to distinguish between conventional, conservative, and zero tillage practices can be challenging without knowing the specific timing of tillage or planting (Zheng et al., 2012, 2013). Therefore, its annual derivation, minimum NDTI (minNDTI), is adopted as a proxy for tillage due to its ease of application (Zheng et al., 2012). We also calculate annual percentile aggregations (25th, 50th and 75th) of NDVI and NDWI to quantify annual vegetation and water conditions. Finally, from these annual time series, we develop long-term indices that reveal decadal characteristics as the 4–tier predictors.

Given the comprehensive and extensive nature of this data cube, it is impractical to validate every derived spectral index against ground truth data. Plausibility checks were conducted selectively, focusing on 3rd tier predictors where feasible and necessary, and where relevant ground data were available. These checks involved matching the data with the ground survey statistics to highlight the advantages and limitations of the data. Our predictors data cube will be compared with EcoDataCube.eu, which has been utilized for land cover (LC) classification by Witjes et al. (2023). By benchmarking against EcoDataCube.eu, our aim is to highlight the improvements and advantages offered by our spectral indices data cube, specifically in terms of accuracy, robustness, and usability for land cover classification. To demonstrate the versatility and utility of the data cube for various environmental modeling purposes, we also present a case study focused on building regression modeling for soil organic carbon (SOC) using thid data cube. Finally, we discuss the recommended uses, limitations, and future development of this data cube and compare it with other similar projects and initiatives. All the layers we produced are available under the open data license CC-BY and can be accessed and visualized via our https://stac.ecodatacube.eu as Cloud-Optimized

GeoTIFFs (Witjes et al., 2023). To enable users to easily assess the suitability of datasets for specific applications, metadata is embedded within the layer name in a structured and standardized format, aligning with Data-Fitness-For-Use assessment methodologies (Yang et al., 2013; Pôças et al., 2014; Lokers et al., 2016; A. Wentz and Shimizu, 2018; Lacagnina et al., 2022). A copy of the data, including bimonthly reflectance bands, spectral indices, annual aggregations andlong-term analysis, is also available from Zenodo (Tian et al., 2024, https://doi.org/10.5281/zenodo.10776891); complete code used to derive all indices is available via https://doi.org/10.5281/zenodo.12907281.

## 2   Material and method

### 2.1   Landsat ARD V2

The data cube presented in this study is based on the Landsat Analysis Ready Dataset version 2 (ARD V2), developed by the Global Land Analysis and Discovery (GLAD) team at the University of Maryland (Potapov et al., 2020). Landsat ARD V2, the second version of Landsat ARD, consists of 16-day tiled composites with 23 images per year from 1997 to 2022, totaling 598 images. It includes seven reflectance bands (blue, green, red, near-infrared, short-wave infrared 1, short-wave infrared 2 and thermal) and a detailed quality flag that classifies each pixel as land, water, cloud, cloud shadow, topographic shadow, hill shade, snow, haze, cloud proximity, shadow proximity, other shadows, or buffered proximity of the previously mentioned classes. These quality flags enable the identification of poor quality pixels, including those affected by clouds, cloud shadows, haze, or other shadowy conditions. The presence of these gaps requires significant preprocessing before the data can be used for direct modeling and analysis. As one of the few globally consistent archives for historical time series of normalized surface reflectances derived from various Landsat satellite collections, Landsat ARD V2 offers long-term availability (since 1997) and detailed spatial resolution (30 meters). For this study, we cropped the global Landsat ARD V2 layers to a pan-European extent, including Ukraine, the UK and Turkey. The GLAD ARD tile system is adopted to support data organization, parallel processing, and spatial inference when necessary.

### 2.2   TSIRF framework and scikit-map

As a comprehensive framework for processing EO time series, Time-Series Iteration-free Reconstruction Framework (TSIRF) enables diverse time-series processing techniques by simply adjusting the convolution kernel (Consoli et al., 2024). We adopted TSIRF for temporal aggregation and gap-filling to impute data gaps created after removing clouds using the ARD 2 quality assessment mask. This framework is implemented through the Python package scikit-map (https://github.com/openlandmap/scikit-map), which optimizes speed by leveraging data structures for parallel computing. In addition to TSIRF, scikit-map also offers straightforward capabilities for band operations, trend calculations, temporal statistics, parallel raster reading and writing, and processing. These functionalities enable efficient processing, analysis, and visualization of large multidimensional raster datasets. In this work, all data processing and map production were performed using scikit-map python library.

## 2.3 Spectral indices predictors data cube

The preparation of the spectral biophysical indices data cube involves developing four tiers of predictors from Landsat ARD V2, step by step. These tiers differ in temporal resolution and processing complexity, as outlined in Fig. 1 and detailed in the following subsections. Section 2.3.1 introduces the production of tier-1 bimonthly, gap-free Landsat reflectance bands. Tier-2 predictors are bimonthly time series for six biophysical indices (Section 2.3.2). From tier-2 indices, time series analyses are conducted to derive annual predictors (Section 2.3.3). Finally, further time series analyses are applied to tier-3 predictors to extract long-term temporal features representing persistent states (Section 2.3.4).

### 2.3.1 First tier of predictors: bimonthly Landsat bands

The first tier predictors are ARCO land surface reflectance bands, gap-filled using Landsat ARD V2 data. Pixels with insufficient quality were identified using the quality band and masked, creating gaps in the time series. These gaps were initially reduced by calculating a weighted average from several scenes (typically 6–7) within and adjacent to the two-month period of the original 16-day interval time series, resulting in a bi-monthly product (one image every two months). Despite this aggregation, significant gaps remained, as shown in Fig. 2, which were subsequently filled using the Seasonally Weighted Average Generalization (SWAG) method developed by (Consoli et al., 2024). SWAG assigns weights for aggregation based on the clear sky fraction of the corresponding tiles. For gap-filling, SWAG respects seasonality and causality by reconstructing each pixel's time series using weighted averages, prioritizing images from the same period in previous years, and relying only on past data.

This Tier 1 product is part of the global bi-monthly aggregation framework established by (Consoli et al., 2024) and represents the European subset of that effort. Within this work, the data has been refined for regional applications by applying the pan-European landmask (https://doi.org/10.5281/zenodo.8171860) to exclude water bodies and assembling it into continental mosaics stored as COGs, as shown in step (1) of Fig. 1. These gap-filled bi-monthly ARCO Landsat surface reflectance layers provide the foundational material for producing spectral indices and their derivatives.

### 2.3.2 Second tier of predictors: bimonthly spectral index

The second tier of predictors is calculated per pixel using ARCO surface reflectance bands, as shown in step (2) of Fig. 1. Direct satellite-derived indices make use of the unique spectral signatures of different objects on the surface of the land to detect their presence within a pixel. Direct satellite-derived indices include Normalized Difference Vegetation Index (NDVI), Soil Adjusted Vegetation Index (SAVI), Fraction of Absorbed Photosynthetically Active Radiation (FAPAR), Normalized Difference Tillage Index (NDTI), Normalized Difference Water Index (NDWI), and Normalized Difference Snow Index (NDSI). The calculation formula we adopted is described in Table 1.

### 2.3.3 Third tier of predictors: annual aggregated index

In step (3), the aggregated annual indices are derived by extracting key temporal features or statistics from the bimonthly index series for each year. These indices are then provided at a 30m spatial resolution on an annual basis:

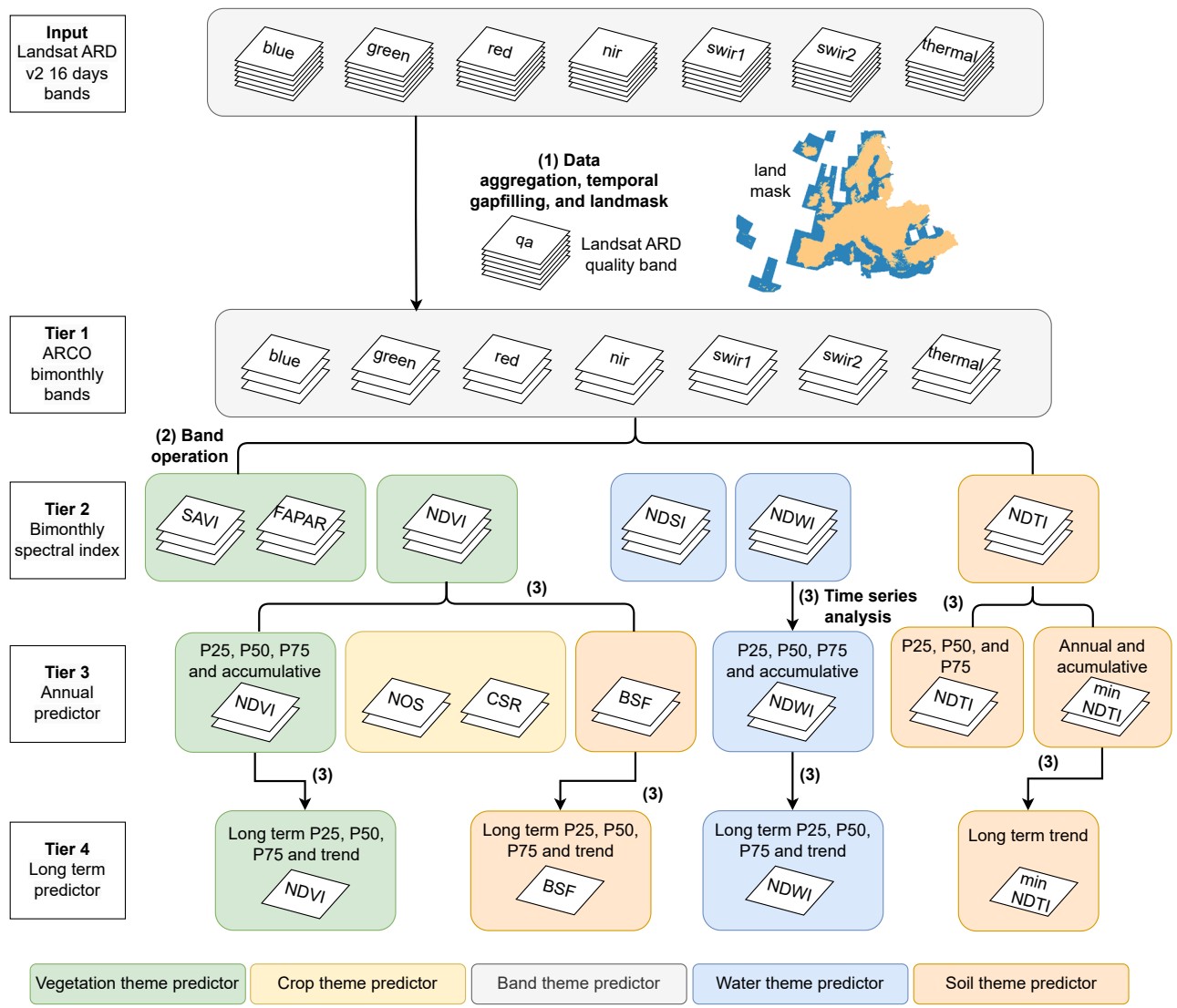

**Figure 1.** General workflow for processing Landsat-based spectral index predictors. They formed four different tiers based on their level of processing and different temporal resolutions through the workflow. The predictors could also be categorized into five thematic groups, each framed by different colors in the figure, including vegetation, water, band-specific properties, crops, and soil characteristics. The "(3) Time series analysis" in the workflow incorporates three different temporal operations: temporal aggregation, which extracts statistical measures over specific periods to represent those intervals; trend analysis, which evaluates the directional changes of a predictor over time; and cumulative analysis, which sum up the corresponding annual predictor values from a starting baseline year (*i.e.* 2000) to the specified index year. Land-mask source: https://doi.org/10.5281/zenodo.8171860.

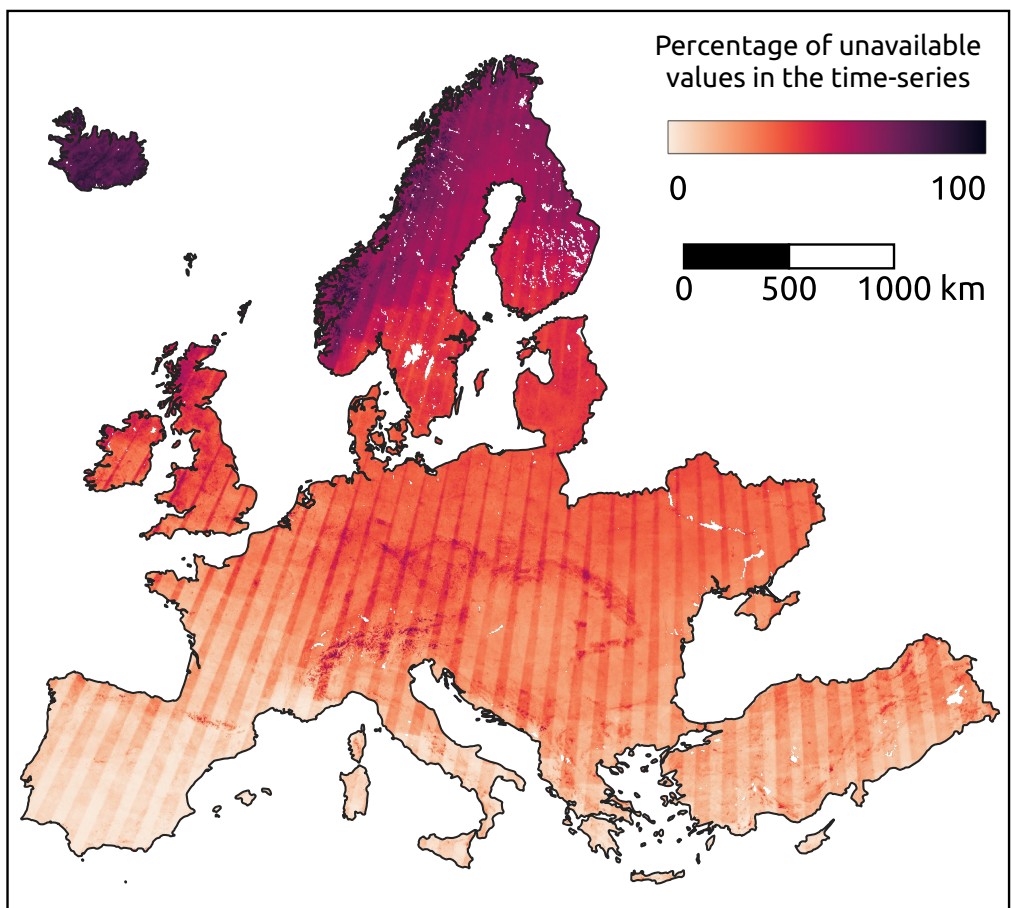

**Figure 2.** Per-pixel count of available values in the bimonthly aggregated Landsat time-series from 1997 to 2022. Darker areas are more affected by presence of data-gaps in the time-series. In addition to cloud-presence and snow-cover, it is possible to notice patterns determined by overlapping scenes in the original Landsat raw images. Adapted from Consoli et al. (2024).

- **Annual P25, P50, and P75 aggregation of NDVI, NDWI and NDTI** are calculated by identifying the values at the 25th, 50th and 75th percentile of the sorted bimonthly NDVI and NDWI values for each pixel within a year.

- **Minimum Normalized Tillage Index (minNDTI)** is determined as the minimum value of the six NDTI values over a year (Zheng et al., 2012).

- **Bare Soil Fraction (BSF)** is calculated by dividing the number of pixels classified as bare surface within a year's time series (identified by the criterion of NDVI values below 0.35) by the total number of pixels analyzed in that year (Castaldi et al., 2019).

- **Number of season (NOS)** indicates the frequency of cropping cycles within a year, calculated by identifying peaks in the NDVI time series (Li et al., 2014; Liu et al., 2020). Time steps with NDVI values greater than 0.5 are first flagged

**Table 1.** Overview of third tier of predictors

| Index | Band Operation | Reference |
|-------|---------------|-----------|
| NDVI | $\dfrac{\mathrm{nir} - \mathrm{red}}{\mathrm{nir} + \mathrm{red}}$ | Tucker (1979) |
| NDTI | $\dfrac{\mathrm{swir1} - \mathrm{swir2}}{\mathrm{swir1} + \mathrm{swir2}}$ | Van Deventer et al. (1997) |
| NDWI | $\dfrac{\mathrm{nir} - \mathrm{swir1}}{\mathrm{nir} + \mathrm{swir1}}$ | Gao (1996) |
| NDSI | $\dfrac{\mathrm{green} - \mathrm{swir1}}{\mathrm{green} + \mathrm{swir1}}$ | Salomonson and Appel (2006) |
| SAVI | $\dfrac{(\mathrm{nir} - \mathrm{red})}{(\mathrm{nir} + \mathrm{red} + 0.5)} \times 1.5$ | Huete (1988) |
| FAPAR | $\dfrac{(\mathrm{ndvi} - 0.03) \times (0.95 - 0.001)}{0.96 - 0.03} + 0.001$ | Robinson et al. (2018) |

as candidate peaks, possibly corresponding to an actual cropping cycle. A prominence filter of 0.25 ensures that each candidate peak is at least 0.25 higher than its surrounding troughs, addressing consistently high NDVI values in areas such as forests. Adjacent peaks with intervals shorter than 2 months are merged, keeping the one with higher NDVI values, to ensure an accurate representation of distinct cropping cycles.

- **Crop Duration Ratio (CDR)** quantifies the length of active cropping periods, calculated as the ratio of time during which a pixel is in an active cropping state, defined as when the vegetation signals reach at least half the amplitude of the phenological curve's peak values (White et al., 1997). The peak values are determined by the average NDVI values of the peaks identified during the calculation of NOS.

- **Accumulated NDVI P50, NDWI P50, NDVI of the month of July and August, and minNDTI** The cumulative indices are calculated by summing their values over time from the year 2000. They are expected to reflect the cumulative effects of water content, vegetation health, and tillage practices on each pixel.

### 2.3.4 Fourth tier of predictors: long-term temporal feature index

In step (4), the indices that quantify long-term temporal features over 2000–2022 are calculated. The long-term changes trends are derived from annual aggregates to represent the changes from 2000 to 2022. For this, we adopted the Theil-Sen estimator to fit the trend for 2000–2022. This is a statistical, non-parametric approach that is resistant to outliers, as it calculates the median slope among all point-pairs (Theil, 1950; Sen, 1968). This results in trend maps for Bare Soil Fraction (BSF), NDVI P50, NDWI P50, and minNDTI. Additionally, the P25, P50, and P75 percentiles of NDVI, NDWI, and BSF are derived for the years 2000-2022 to provide a general overview of their distribution and general state over the this period.

## 2.4 Plausibility check

Due to the limited availability of land survey data that align temporally, spatially, and thematically with our data cube, we were unable to perform traditional accuracy validation on most index layers, which requires extensive ground-truth datasets for direct comparison. Instead, we conducted a plausibility check to assess whether the data align logically and statistically with established land surface processes. Apart from tier 1 product, we focus our quality assessment on the third tier of predictors—annual aggregated indices such as minNDTI, BSF, NOS, and CDR—since they are more complex and less established than simpler spectral indices like NDVI and NDWI.

### 2.4.1 Quality assessment of tier-1 product

The quality of the gap-filled time series for bimonthly Landsat surface reflectance bands was evaluated using 2,746 time series randomly sampled from Europe. Details are provided in the supplemental computational notebook available in the Section 7. To simulate real-world conditions, 10% artificial gaps were introduced into the data. Performance metrics were used to assess the accuracy of the TSIRF model, including R-squared (R2) which measures the proportion of variance explained by the model; Root Mean Squared Error (RMSE), quantifying the average magnitude of prediction errors; and Concordance Correlation Coefficient (CCC; Lawrence and Lin 1989), evaluating the agreement between observed and predicted values are reported to indicate the accuracy of TSIRF in reconstructing the tier-1 land surface reflectance bands.

### 2.4.2 Index statistics by crop type: BSF, NOS and CDR

To examine the effectiveness of crop-specific parameters — BSF, NOS and CDR-in reflecting the crop patterns — we used the harmonized LUCAS (Land Use and Cover Area frame Survey) data, which details crop types information at specific locations and times (d'Andrimont et al., 2020). Three NUTS2 regions are selected to capture a diverse array of European agricultural practices and climates: Picardy region (FRE2), Piedmont province (ITC1), and Malopolska Province (PL21). Picardy is chosen for its temperate maritime climate and highly mechanized, large-scale farms. Piedmont is included to represent its unique damp rice paddies, which illustrate specialized crop cultivation. Malopolska offers insights into agriculture under a more continental climate, contrasting with other regions. We identified points belonging to four predominantly grown crops in the selected NUTS2 regions—FRE2, ITC1, and PL21 from the most recent LUCAS 2018 dataset. These points were then overlaid onto

the 2018 index maps for BSF, NOS and CDR, followed by the calculation of the statistics for crop-specific indices among the different crop types. By comparing the statistics of these indices across regions and crops, we conducted an evaluation of the indices' sensitivity and accuracy in capturing the distinctions in cropping patterns and reflecting the specific agricultural dynamics.

### 2.4.3 Correlation with land survey data: BSF

We matched BSF maps against two land survey datasets, with approximately 150 records of crop cover duration from 2007 to 2016 on cropland (Edlinger et al., 2022, 2023). The hypothesis is that the proportion of time that crops cover the soil is inversely related to the presence of bare soil during the same time period. By implementing a correlation analysis between BSF values derived from satellite data and ground-based measurements of crop cover duration, our objective was to assess the reliability of the BSF in capturing the extent of bare soil across agricultural landscapes.

### 2.4.4 Correlation with regional statistics: minNDTI

To validate the minNDTI's effectiveness in reflecting tillage practices across the EU, we compared it with Eurostat's tillage area statistics (https://doi.org/10.2908/EF_MP_PRAC). Eurostat's data originates from farm structure surveys conducted in 28 EU countries and is available in a vector shapefile format covering 319 NUTS2 regions, detailing arable areas under different tillage practices. The breakdown of arable land area based on tillage practices is as follows:

$$\sum(\text{TIL\_CV} + \text{TIL\_CSERV} + \text{TIL\_ZERO} + \text{ARAXTIL}) = \text{ARA} \tag{1}$$

where TIL_CV refers to areas under conventional tillage, TIL_CSERV refers to areas under conservation tillage, TIL_ZERO refers to areas under zero tillage, ARA refers to total arable land areas, and ARAXTIL refers to areas land excluding from tillage survey; all areas recorded in hectares. Notably, ARAXTIL represents land that is excluded from tillage survey due to not being sown/cultivated during the reference year, which includes areas with multi-annual plants such as temporary grassland, leguminous plants, industrial crops (hops or aromatic plants) etc. These are recorded in hectares and together make up the total arable land area (ARA).

To spatially match the Eurostats data with our predictor layers, the data preparation process involved four steps:

1. **Calculate Tillage Shares**: For valid NUTS2 data records with all four types of tillage practices, calculate the share of each tillage practice within each NUTS2 region.

2. **Crop Masking**: Remove non-arable pixels from the minNDTI maps using the EU crop mask developed by d'Andrimont et al. (2021).

3. **Spatial Overlay**: Overlay the Eurostat survey data with crop-masked minNDTI map layers of matching years.

4. **Calculate Average minNDTI**: Calculate the average minNDTI value within each NUTS2 region.

The relationship between each tillage practice and minNDTI is assessed separately using Ordinary Least Squares (OLS) regression models, correlating the shares of each tillage practice with the average minNDTI values within each NUTS2 region. To capture the collective impact of all tillage practices, we introduced the concept of Weighted Crop Residue Cover (WCRC). This metric aggregates the influence of various tillage practices, considering minNDTI's correlation with crop residue cover (CRC):

$$WCRC = \sum_{\text{tillage types}} (\text{typical CRC value} \times \text{area share}) \tag{2}$$

where WCRC integrates typical CRC values for each tillage type, weighted by their respective area shares within each NUTS2 region.

We fitted typical CRC values for each type of tillage practice from their respective ranges, as outlined in Table 2 (Magdoff et al., 2000; Zheng et al., 2012). This optimization process identified the values within these ranges that maximize the correlation between WCRC and minNDTI. For excluded tillage practices, we assume a broad range of 0 to 100 percent to cover all possibilities, reflecting the uncertainty in its typical CRC values. The correlation analysis assesses the relationship between the estimated WCRC and the average minNDTI of the NUTS2 regions.

**Table 2.** Typical CRC value range for each tillage practice type

| Tillage Practice Type | CRC Range (%) |
| --- | --- |
| Conventional tillage | 0–30 |
| Conservation tillage | 30–70 |
| Zero tillage | 70–100 |
| Excluded tillage practices | 0–100 |

## 2.5 Comparison to EcoDataCube in LC classification

The Landsat-based predictor data cube are intended as an improvement of the version in the EcoDataCube (Witjes et al., 2023). The switch to a bimonthly temporal aggregation from a seasonal one with three percentiles leads to a smaller number of variables (6 instead of 12) and a higher level of temporal resolution. In order to quantify the difference of these aggregation techniques, we compare the performance of sets of random forest classifiers trained on both sets of Landsat data to predict seven types of land cover: Cropland, Grassland, Bare land, Forest, Artificial land, Shrubland, and Wetlands. We exclude the EcoDataCube non-Landsat data (DTM and Sentinel–2) and indices in this work to guarantee an objective comparison. Details about the training data will be discussed in the next section, while an overview of the experiment workflow is presented in Fig. 3.

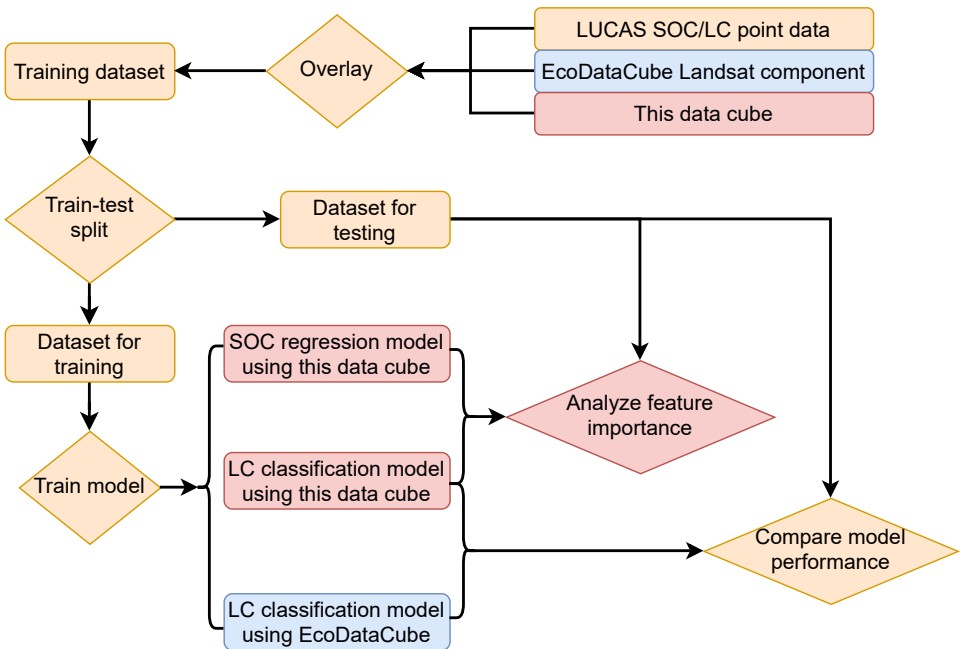

**Figure 3.** Workflow for modeling experiments, illustrating the comparison between EcoDataCube and the proposed data cube in the LC classification experiment, as well as the feature importance analysis conducted for the LC classification model and the SOC regression model using the proposed data cube. Components specific to EcoDataCube are marked in blue, those unique to the data cube produced in this work are marked in red, and shared general components are marked in yellow.

## 2.6 Modeling experiments

To demonstrate the utility of the indices in the data cube, we performed two modeling experiments: soil organic carbon (SOC) regression and LC classification. The general workflow is depicted in Fig. 3. The data points used for SOC regression and land cover classification come from the LUCAS land cover and top soil surveys. Only surveyed point data (*i.e.* identified by unique point id and sampling year) with available both land cover and topsoil SOC data were selected for modeling experiments. These point data were overlaid with EcoDataCube Landsat data, along with all Landsat reflectance and biophysical index layers generated in this study, to create the training dataset for modeling experiments. Using stratified sampling, we select about 10 % of test data points based on a 30 km grid tiling system to ensure that the test data accurately represent the spatial distribution throughout the study area. This results in 52.306 training data points and 8.417 test data points, in total 60.723 points.

The models' performance for both tasks is evaluated by training them with different combinations of predictors, organized into two main categories: themes and tiers. The first division categorizes the predictors by themes: reflectance bands, vege-

tation, water, soil, and crop (see Table 3). The second division organizes predictors by tiers, which represent different levels of temporal aggregation and processing, as detailed in Section. 2.3. These tiers, as described above, include bimonthly bands, bimonthly indices, annual indices, and long-term indices. By systematically testing models with these combinations, we can understand how indices developed for different purposes and how short-term variations, seasonal dynamics, and long-term

temporal features contribute to the accuracy and robustness of predictive models. Both SOC regression and land cover classification models were trained using different groups from both divisions. We examined the feature space to identify the most influential predictors and compared their importance between the regression and classification tasks. This analysis aimed to highlight the flexibility and comprehensive nature of our data cube. Apart from comparing different feature combinations, we also provide the top 20 most important features for the model trained on all predictors and all point data. This provides a

focused analysis that identifies which specific predictors are the most influential across the entire dataset when no subsets are used.

**Table 3.** Theme-Based Division of Predictors

| Theme | Predictors |
|---|---|
| Reflectance Bands | Bimonthly Landsat bands |
| | Annual P25, P50, and P75 of Landsat bands |
| Vegetation | Bimonthly NDVI, SAVI, FAPAR |
| | Annual NDVI P25, P50, and P75; acumulated NDVI P50 and NDVI of the bimonth July and August |
| | Long term P25, P50, P75 and trend of NDVI |
| Water | Bimonthly NDWI, NDSI |
| | Annual NDWI P25, P50, and P75 |
| | Long term P25, P50, P75 and trend OF NDWI |
| Soil | Bimonthly NDTI |
| | Annual minNDTI, NDTI P25, P50, and P75, and BSF |
| | Long term P25, P50, P75 and trend of BSF, and long term trend of minNDTI |
| Crop | Annual NOS and CDR |

For the SOC regression experiment, we used the random forest (RF) as an estimator to predict the SOC content throughout the EU. The target variable, SOC, was obtained from the LUCAS soil survey. We evaluated the performance of the regression model using different combinations of predictors to identify which set provided the most accurate predictions. Performance

metrics used to assess model accuracy include R2, RMSE, and CCC. We used a Random Forest regression model with the following parameters: a maximum depth of 20 to control complexity, `sqrt` for max features to improve performance, a minimum of 2 samples per leaf to prevent overfitting, a minimum of 5 samples per split for stable node splitting, and 800 estimators to

ensure robust predictions. These parameters were selected through empirical testing and cross-validation, implemented using the scikit-learn library in Python. The metrics will also be calculated in the *log1p* space.

For the LC classification experiment, we employed a random forest classifier to predict LUCAS LC classes. Similarly to the SOC regression, we tested the model's performance with different combinations of predictors to determine the most effective feature sets. Performance metrics used to evaluate the classification model include precision, assessing the accuracy of positive identifications; recall, evaluating the model's ability to capture all relevant cases; and F1-score (Taha and Hanbury, 2015), providing a balance between precision and recall for a comprehensive model assessment.

## 3 Result

This section begins with an accuracy assessment of time series reconstruction using the TSIRF method to generate tier-1 predictors. This is followed by a qualitative visual examination of spatial patterns in three zoomed-in areas, each representing distinct dominant land cover types and unique land dynamics over the past decades. Additionally, long-term trends of BSF, NDVI, and NDWI are analyzed using annual time-series data from 2000 to 2022 at three LUCAS points within these areas. These analyses provide both spatial and temporal insights into land cover changes and agricultural practices, demonstrating how these indices capture and reflect patterns and trends over time. Where relevant data are available, quantitative assessments are also incorporated to further validate our approach, as elaborated in the Section 2.4.

### 3.1 Accuracy of reconstructed tier-1 predictors

The Table 4 presents performance metrics for the reconstructed Landsat surface reflectance bands across tier-1 seven spectral bands. Overall, the reconstruction demonstrates high accuracy, with low RMSE, high $R^2$ and CCC across all bands. Thermal band consistently shows the best performance, while the NIR band exhibits the lowest performance, with slight variations observed among the other bands. In a similar global-scale experiment, Consoli et al. (2024) reported generally higher reconstruction performance. For a more comprehensive perspective on global time-series reconstruction, we refer readers to their work, which focuses exclusively on this topic.

**Table 4.** Performance metrics for the reconstruction of tier-1 product

| Metric | Blue | Green | Red | NIR | SWIR1 | SWIR2 | Thermal |
|---|---|---|---|---|---|---|---|
| **RMSE** | 0.04 | 0.04 | 0.04 | 0.05 | 0.03 | 0.02 | 0.01 |
| **RMSE/$\mu$** | 0.86 | 0.51 | 0.56 | 0.20 | 0.15 | 0.22 | 0.01 |
| **RMSE/$\sigma$** | 0.56 | 0.56 | 0.58 | 0.60 | 0.41 | 0.40 | 0.32 |
| **$R^2$** | 0.67 | 0.67 | 0.67 | 0.62 | 0.83 | 0.84 | 0.90 |
| **CCC** | 0.83 | 0.82 | 0.81 | 0.79 | 0.91 | 0.91 | 0.95 |

## 3.2 Visual examination of tier-3 predictors

Figure 4 presents zoomed-in examples of tier-3 predictors: NDVI P50, NDWI P50, BSF, NOS, and CDR. They provide an overview of vegetation health, moisture levels, bare soil exposure, number of growing seasons, and crop duration. These visual representations complement the statistical analysis by highlighting spatial patterns that may not be evident through quantitative methods alone, providing an intuitive understanding of the spatial patterns of the indices. The three presented regions have distinct features (Fig. 4): (A) an agricultural area in northern France, characterized by well-organized grid-like field patches of various crops; (B) a rice cultivation region that featured less regular field texture in northern Italy; and (C) the former Szczakowa sand mine in southern Poland (Pietrzykowski and Krzaklewski, 2007; Pietrzykowski, 2008).

NDVI and NDWI delineate agricultural patches with 30 m spatial resolution (Fig. 4-A). Forested areas show high values of NDVI and NDWI, while urban areas (scattered towns in Fig. 4-A and Fig. 4-B) and exposed land from mining activities (Fig. 4-C) exhibit low values. High NDWI values in region B, attributed to flooded paddy fields, do not always coincide with high NDVI (Ranghetti et al., 2016). For example, there are several patches in Fig. 4-B exhibiting high NDWI but moderate NDVI values. This observation could be attributed to the paddy fields in region B, which are flooded in a controlled way to meet the water demand of the rice plants (Ranghetti et al., 2016).

Although derived from the NDVI annual time series, the BSF shows different patterns. Urban and exposed soil areas have high BSF values, while woodlands have low values. Despite lower NDVI P50 values, region B has lower BSF values due to stable NDVI levels above a threshold, indicating less bare soil exposure throughout the year. This highlights that BSF reveals different aspects of the landscape compared to NDVI P50, making it a valuable complement to assess exposure to bare soil and vegetation dynamics. The crop-specific NOS and CDR provide insights into growing seasons and their duration. CDR highlights the duration for which crops occupy a pixel within a year and generally inversely relates to BSF. Most croplands in regions A and B have a NOS value of 1, indicating a single annual growing season, while urban and forested areas have a NOS value of 0 (Fig. 4-C). Some areas show NOS values of 2, indicating possible practices such as winter cropping or double-cropping systems.

## 3.3 Trend analysis with land cover dynamics

Four trend maps were produced to analyze land surface trends between 2000 and 2022: NDVI annual median, NDWI annual median, BSF, and minNDTI (Fig. 5). The NDVI trends reveal a positive trend in most parts of Europe. However, specific regions, including the Alps, Northern Europe, northern Scotland, Iceland, and Scandinavia, exhibit clear negative NDVI trends. These areas, characterized by high latitudes or altitudes, often experience persistent snow cover and frequent cloudiness. The NDWI trend map for these regions shows a positive trend, indicating an increase in vegetation water content. In other parts of Europe, the NDWI trends display a mix of positive and negative changes, lacking a definitive overall trend. The decreasing trend in NDVI in regions of high altitude and high latitude is also reflected in the BSF trend map (Fig. 6). In Spain and Turkey, strong positive and negative trends are scattered, with the positive trend being more dominant. Generally, our results show an

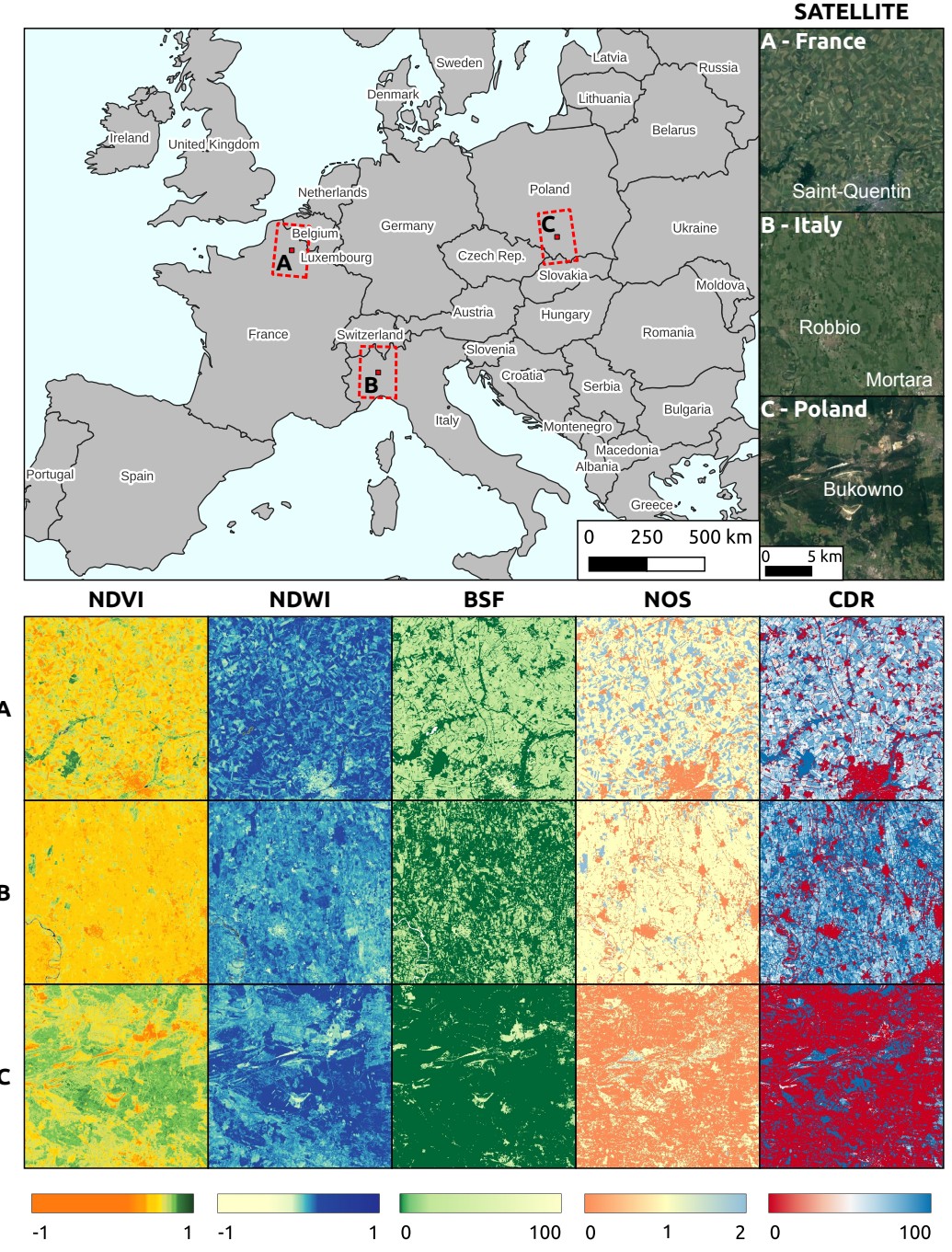

**Figure 4.** Zoomed-in examples of NDVI and NDWI median in the year 2018 at (A) Northern France, with the city of Saint-Quentin at south-east corner; (B) Northern Italy, in the western Po Valley, east of Vercelli; (C), Southern Poland, within the Przemsza River basin, featuring the Szczakowa sand mine. Each image tile has a side length of 20km. Satellite imagery source: © Google Earth (https://earth.google.com/web/).

increase in minNDTI from 2000 to 2022 across Europe. However, there are exceptions, such as the negative trends observed in regions such as Turkey.

In Region A, Northern France, all BSF, NDVI, and NDWI trends show scattered increases and decreases, likely influenced by well-planned crop patches and their varying management practices (Fig. 7). Most areas exhibit positive NDVI and NDWI trends, except for urban areas such as Saint-Quentin city, where both vegetation and water content are declining. The time series also shows significant variation, influenced by crop types and management practices.

In Region B, northern Italy, the NDVI trends exhibit variability, but the overall trend is not as positive as in Region A (Fig. 8). BSF generally remains stable, while some areas exhibit an increasing trend, indicating greater soil exposure. The dominant decline in NDWI highlights a drying trend, likely due to the increasing adoption of dry seeding practices in rice cultivation (Ranghetti et al., 2016). The time series for the three indices has remained relatively stable over 20 years, reflecting consistent rice cultivation practices.

Region C in Poland, which has transitioned from a sand mine to a reforested area (Pietrzykowski and Krzaklewski, 2007; Pietrzykowski, 2008), shows the most significant positive trends in BSF, NDVI, and NDWI (Fig. 9). This indicates a steady increase in vegetation cover and water content, which is also reflected in the time series data. Although it was a coniferous forest from 2009 to 2018, the increases in NDVI and NDWI, coupled with the decrease in BSF, are likely due to the continuous growth of trees. Fig. 10 shows satellite images from 2000, 2010, and 2020, each accompanied by the corresponding BSF images for comparison, clearly illustrating the progression of reforestation during the last two decades.

## 3.4 Crop specific indices statistics across European regions

Fig. 11 shows the average values of BSF, NOS, and CDR for maize, sugar beet, rice, which are common crops in the NUTS2 regions PL21, FRE2, and ITC1, respectively. The NDVI time series for 2018, from which these indices are derived, are shown in the bottom panel. Sugar beet in the FRE2 region has the highest average NOS value, due to its distinct NDVI time series fluctuations, indicating pronounced seasonal variations. The NDVI time series for sugar beet shows a distinct peak during the summer months (July and August), a deep valley in March and April, and a less pronounced peak in winter. In comparison, the NDVI time series for maize in the PL21 region and rice in the ITC1 region display more uniform patterns, with smaller differences between peak and valley NDVI values, leading to lower NOS values. Sugar beet in the FRE2 region also exhibits the highest BSF values due to a long and deep valley period in the NDVI time series. In contrast, rice in the ITC1 region shows the lowest BSF values, reflecting its stable vegetation coverage throughout the year. Consequently, rice in the ITC1 region also has the highest CDR value. Although BSF and CDR often show a negative correlation, this is not always the case. For example, sugar beet in the FRE2 region has a higher BSF compared to maize in the PL21 region but also has a higher CDR. This highlights that, although related, BSF and CDR can differ due to their different focus: BSF measures the duration of bare soil, whereas CDR emphasizes the active growing period.

Fig. 12 shifts the focus from different crop types across regions to the same crop type—common wheat—grown in the FRE2, PL21, and ITC1 regions. The average values of BSF, NOS, and CDR for common wheat reveal notable regional variations, despite the NDVI time series overlap. Common wheat in the FRE2 region displays significantly higher NOS values compared

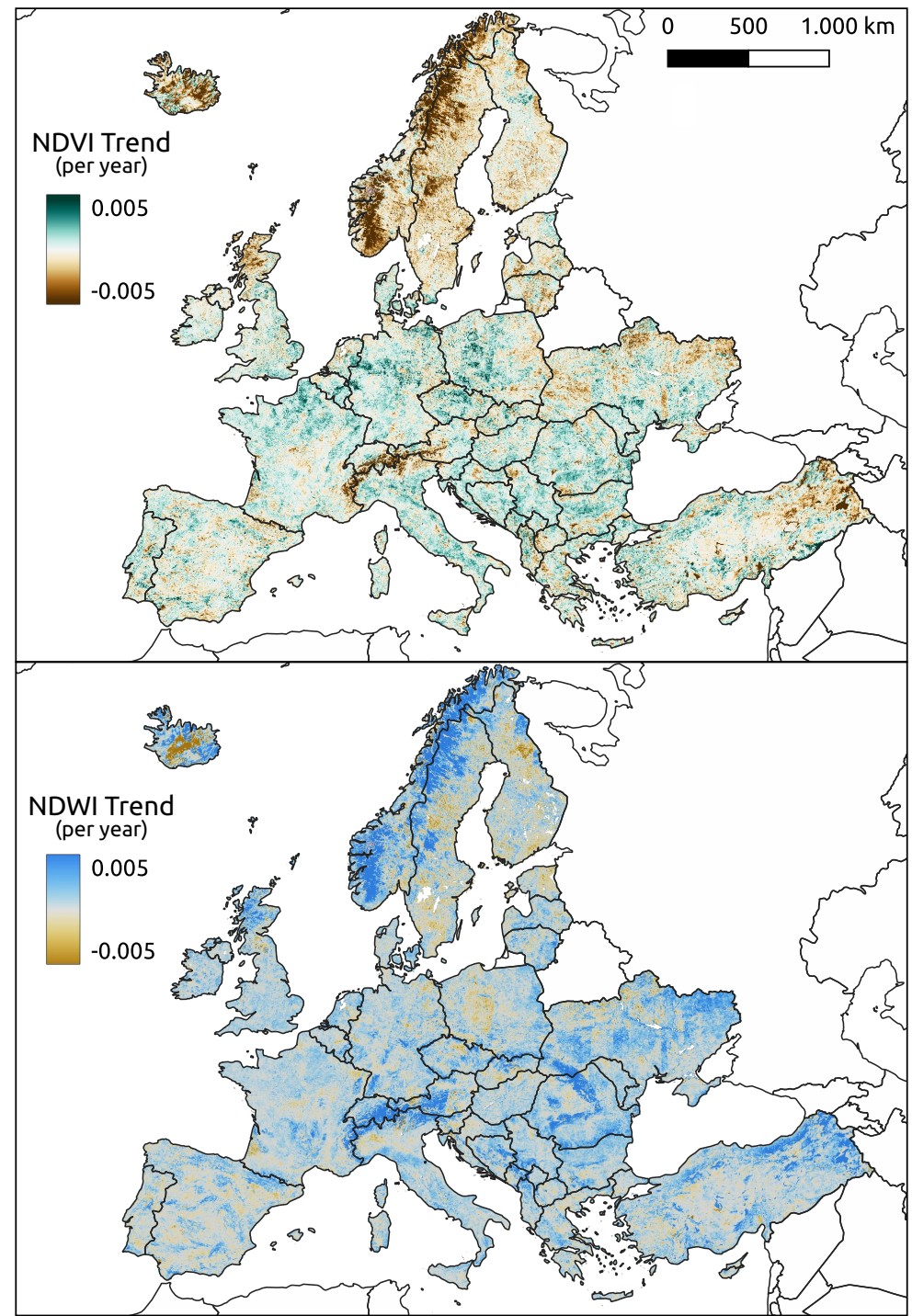

**Figure 5.** Long term changing trend between 2000–2022 of NDVI (top panel) and NDWI (bottom panel), as the annual rate of change in index values.

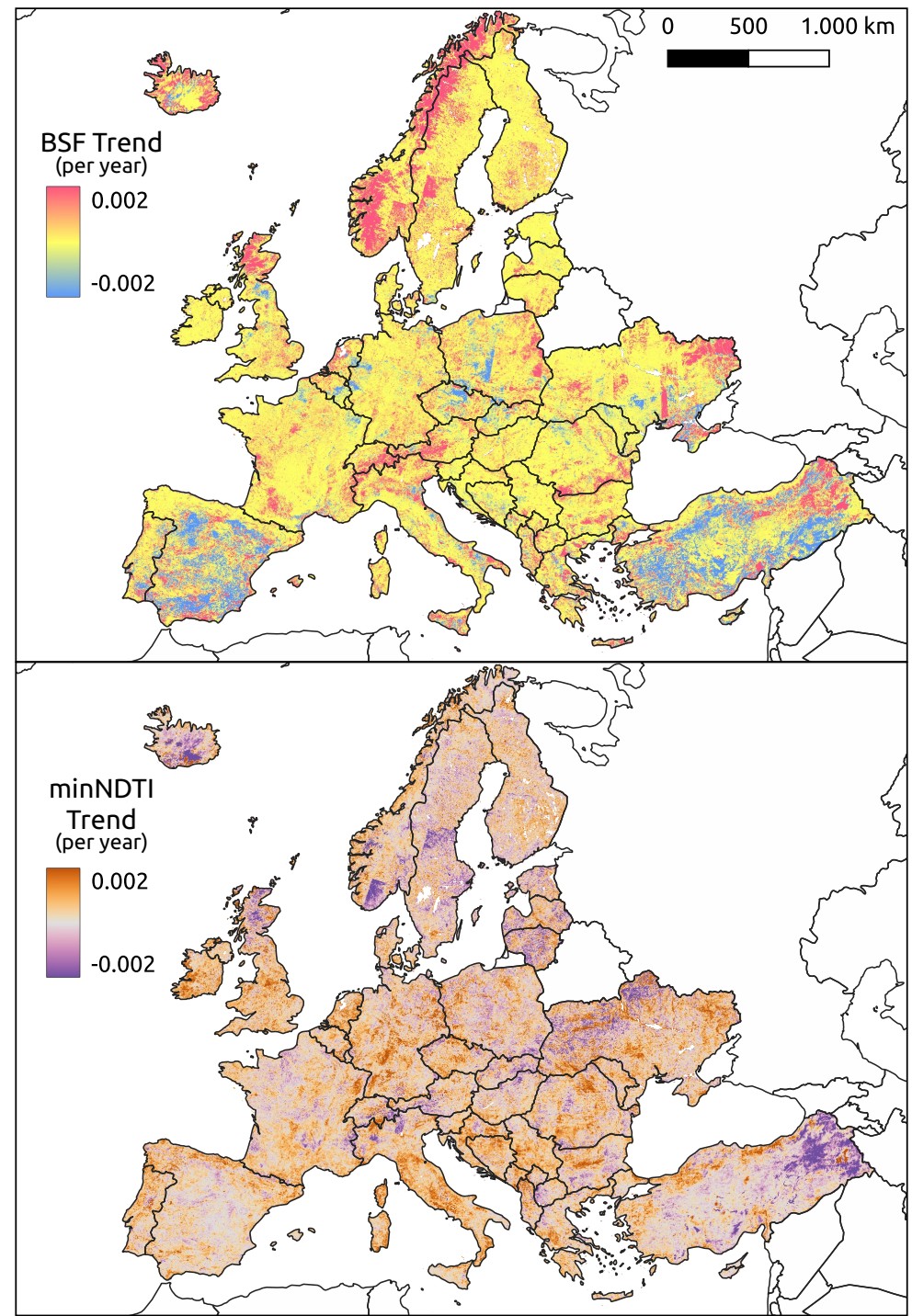

**Figure 6.** Long term changing trend between 2000–2022 of BSF (top panel) and minNDTI (bottom panel), as the annual rate of change in index values.

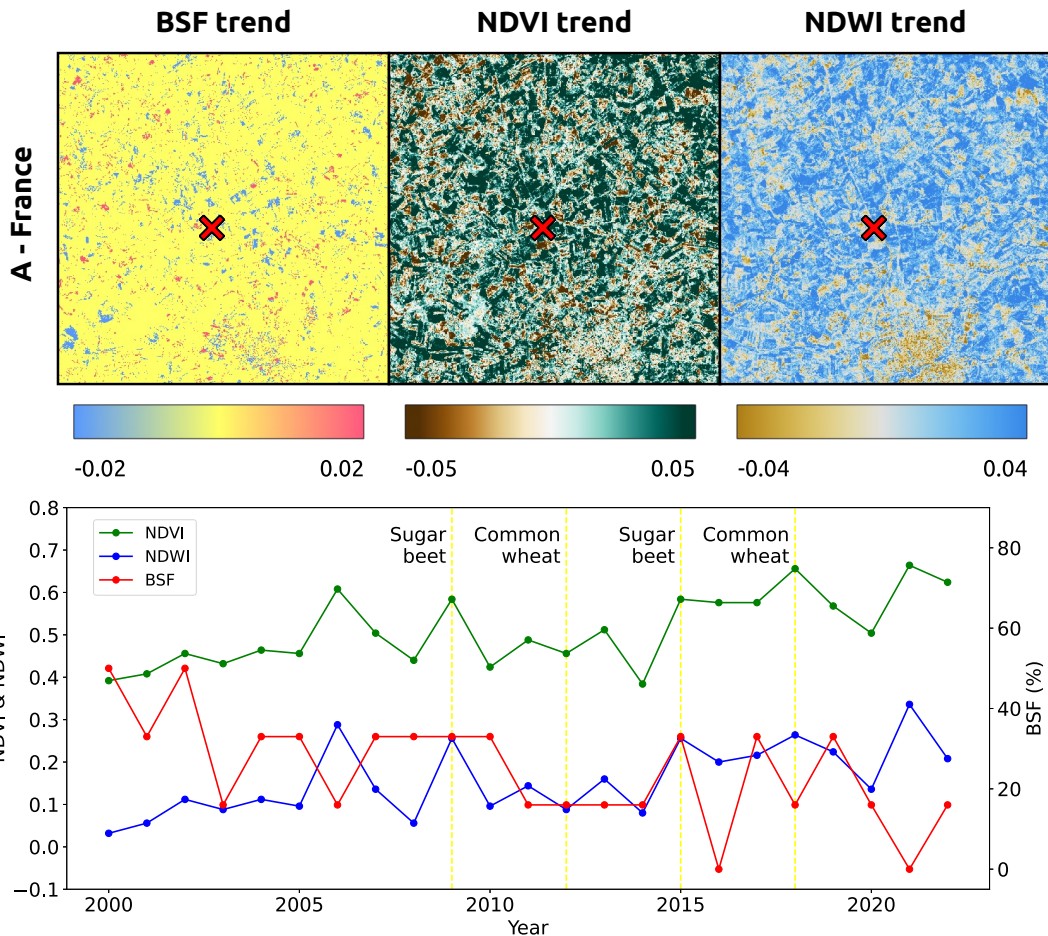

**Figure 7.** The top panel provides a zoomed-in view of NDVI, NDWI and BSF trends for Saint-Quentin region, north France (region A, identical in spatial location and extent to region-A of Fig. 4), which has a consistent agricultural use with varies crop types across years. Images generated by the authors using data produced in this study. Red crosses in the top panel mark the LUCAS point (ID: 38363000; lat: 49.91122N, lon: 3.237571E), detailed in the bottom panel. The bottom panel shows the time series of NDVI, NDWI, and BSF from 2000 to 2022 at this LUCAS point, which is revisited four times in the years 2009, 2012, 2015, and 2018, with the surveyed land cover indicated beside yellow vertical lines.

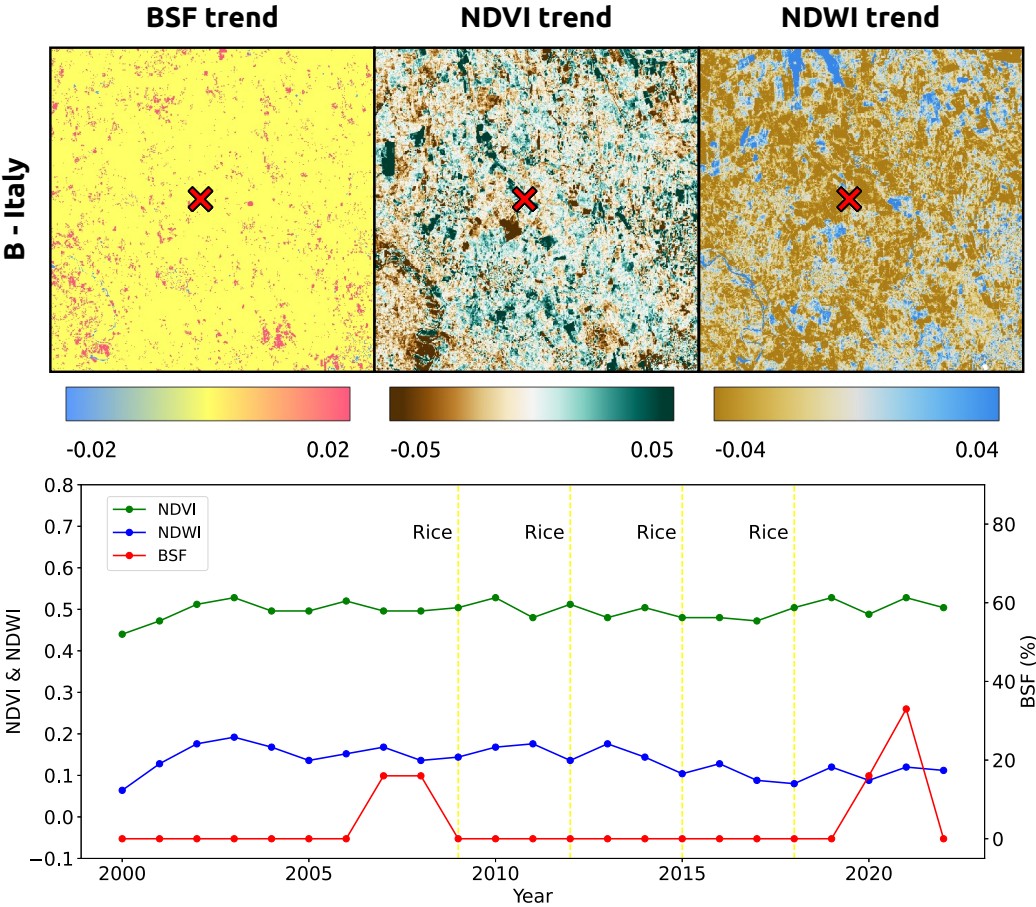

**Figure 8.** The top panel provides zoomed-in view of NDVI, NDWI and BSF trends for rural area besides city Vercelli in north Italy, with dominant rice cultivation (region B, identical in spatial location and extent to region-B of Fig. 4). Images generated by the authors using data produced in this study. Red crosses in the top panel mark the LUCAS point (ID = 42122479; lat=45.33338N, lon=8.611634E), detailed in the bottom panel. The bottom panel shows the time series of NDVI, NDWI, and BSF from 2000 to 2022 at this LUCAS point, which is revisited four times in the years 2009, 2012, 2015, and 2018, with the surveyed land cover indicated beside yellow vertical lines.

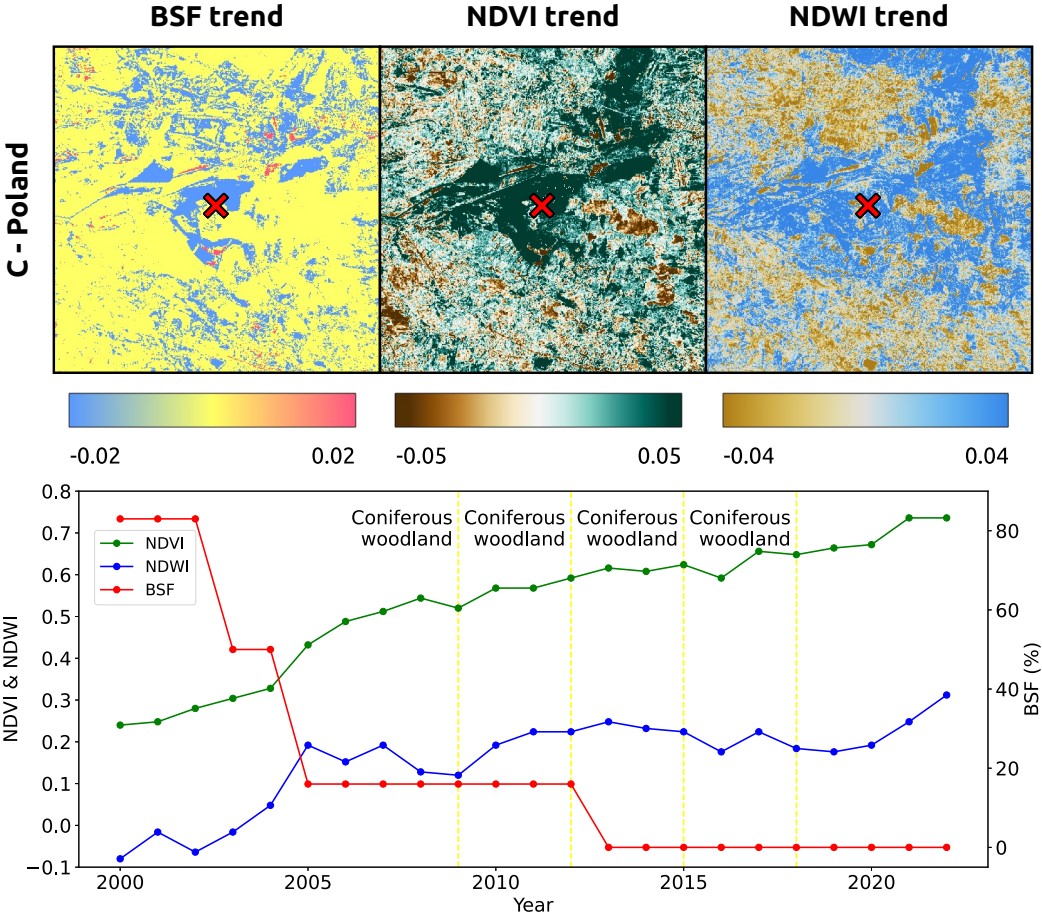

**Figure 9.** The top panel provides zoomed-in view of NDVI, NDWI and BSF trends for the former Szczakowa sand mine, south Poland (region C, identical in spatial location and extent to region-C of Fig. 4). Images generated by the authors using data produced in this study. Red crosses in the top panel mark the LUCAS point, detailed in the bottom panel (ID=49923058; lat=50.24478N, lon=19.435963E). The bottom panel shows the time series of NDVI, NDWI, and BSF from 2000 to 2022 at this LUCAS point, which is revisited four times in the years 2009, 2012, 2015, and 2018, with the surveyed land cover indicated beside yellow vertical lines.

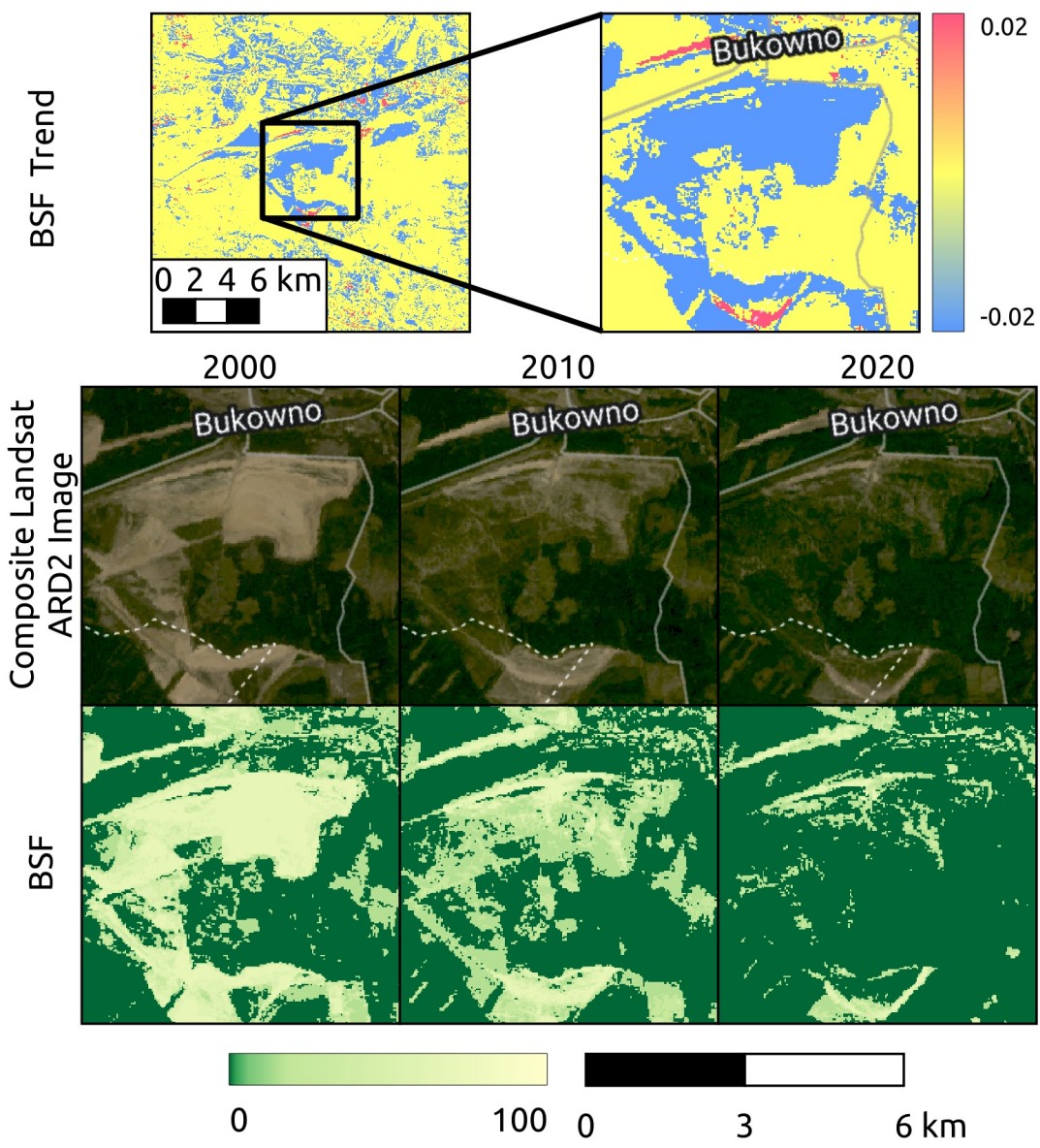

**Figure 10.** Time series comparison showing zoomed-in satellite and BSF images of reforested post-sandmine areas near Bukowno town for the years 2000, 2010, and 2020. On the top left are the BSF trend image, the black frame shows the zoomed-in area of which the satellite and BSF images are shown. The top left image is the same as in Fig. 9. BSF images generated by the authors using data produced in this study.The Composite Landsat ARD2 Image was generated by the authors using data from (Potapov et al., 2020), as described in Sect 2.1.

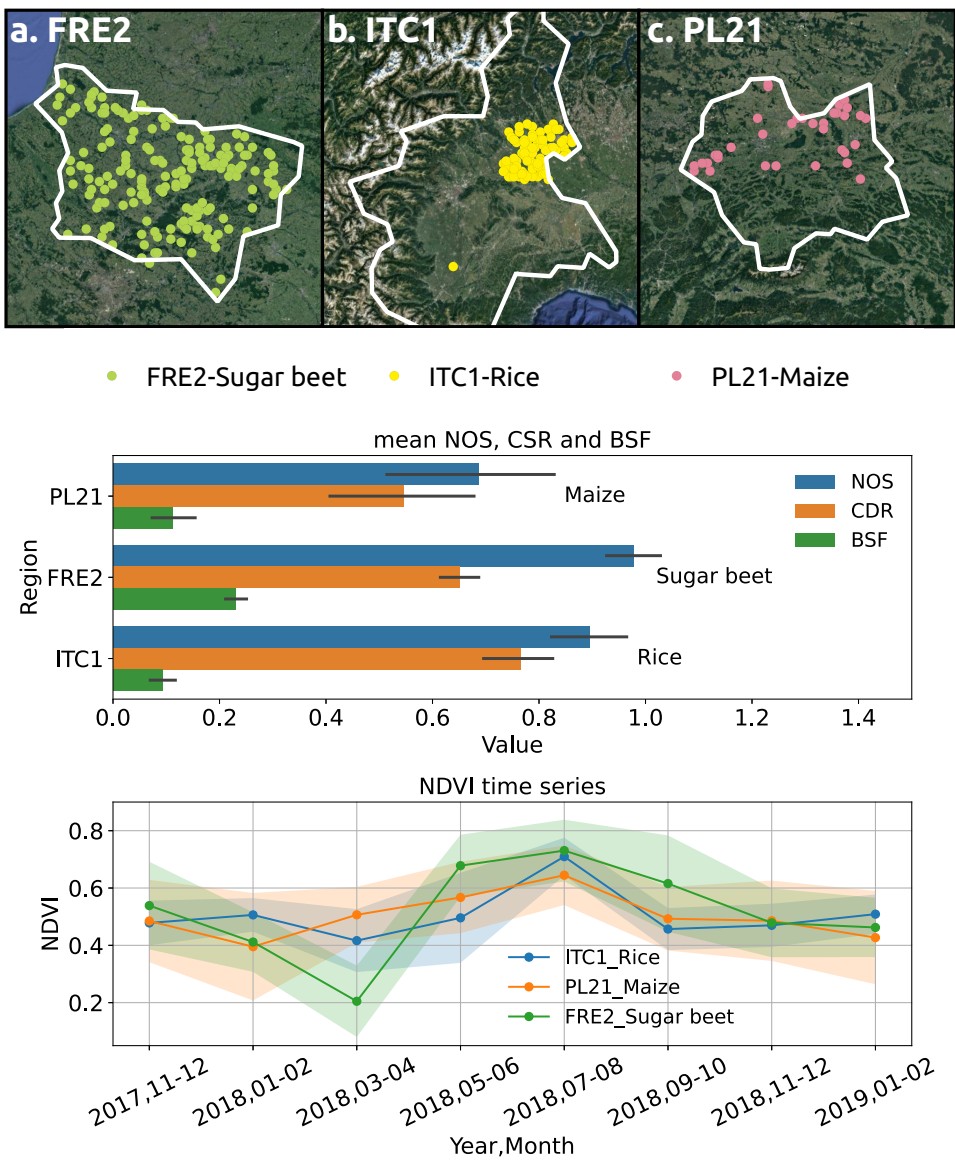

**Figure 11.** Illustration of the locations of LUCAS 2018 survey points of selected crop types (top panel); statistics of various indices calculated from these LUCAS 2018 points across crops in the selected regions (middle panel); and average NDVI dynamics for the year 2018, including adjacent values from neighbouring years (bottom panel). The LUCAS 2018 points data are from 3 NUTS2 regions: (a) the central part of the Picardy region (FRE2) in north France; (b) the western part of the Piedmont province (ITC1) in Italy; and (c) the Malopolska Province (PL21) in southern Poland. These 3 regions are close to or partially overlap with the areas shown in Fig. 4, thus they are denoted with lowercase letters instead of capital ones to show the difference and similarity. Satellite imagery source: © Google Earth (https://earth.google.com/web/).

to the other two regions, due to a secondary growth peak in winter. It also has the highest CDR value, indicating the most active growing cycles. In contrast, common wheat in the ITC1 region has the lowest BSF.

## 3.5 Assessing BSF in cropland

An inverse correlation, as illustrated in Fig. 13, is evident between the recorded crop coverage data and the BSF values from 2007 to 2016 in the cropland. Given the complexities involved, including the varying temporal resolutions of observation, the effects of crop residues, and the emergence of natural vegetation and weeds, a perfect one-to-one correlation between these variables is not expected. This observed correlation, in line with our hypothesis, confirms the validity of the BSF data.

## 3.6 Compare minNDTI with Eurostat at NUTS2 level

Fig. 14 shows the Pearson correlation between the shares of each individual type of tillage practice and their impact on the minNDTI values. Although the correlation coefficients for all the tillage practices suggest only a moderate correlation between the mean minNDTI values and the respective shares of each tillage practice, comparing the regression coefficients provides valuable information about their relative impacts. The sensitivity of the average minNDTI to each tillage practice is represented by the regression slope coefficient obtained from the OLS model fitting. Conventional tillage practices, characterized by soil inversion and minimal soil cover post-harvest, exhibit the most negative influence on average minNDTI in NUTS2 regions. This is followed by conservation tillage practices. In contrast, zero-tillage practices, which minimize soil disturbance, show the most positive effect on the average minNDTI. The area excluded from the tillage practices analysis, denoted as *"araxtil"*, consists mainly of multiannual plants, resulting in a positive impact on the average minNDTI as indicated by the sensitivity analysis.

The correlation analysis between the WCRC and the average minNDTI of NUTS2 regions indicates that the cumulative impact of all tillage practices on the average minNDTI is more significant than any individual practice (Fig. 15). The optimal CRC for the tillage practices excluded from the analysis is relatively high at 75.08 %, consistent with Eurostat's statement that the majority of land excluded from tillage practices comprises multi-annual plants.

To further illustrate the spatial distribution and temporal trends of minNDTI, we used the minNDTI map, masked by the crop mask of d'Andrimont et al. (2021), to generate maps showing the average minNDTI for 2016 across the croplands of each NUTS2 region. Furthermore, we created a map that illustrates the average minNDTI trend between 2000 and 2022 per NUTS2 region (Fig. 16). In 2016, NUTS2 regions in Scandinavia and Ireland exhibited high minNDTI values, in contrast to regions in Spain and eastern Europe, which generally displayed lower minNDTI values on average. The trend map for minNDTI reveals that NUTS2 regions around the Baltic Sea are experiencing a relatively strong negative trend, indicating a possible increase in the intensity of tillage. In contrast, areas in eastern Britain, the Netherlands and Austria exhibit a moderate positive trend. In general, the European Union presents a negative trend in minNDTI. However, caution must be exercised when interpreting minNDTI data, as the correlation between minNDTI values and tillage practices is not robust enough to draw definitive conclusions without additional information.

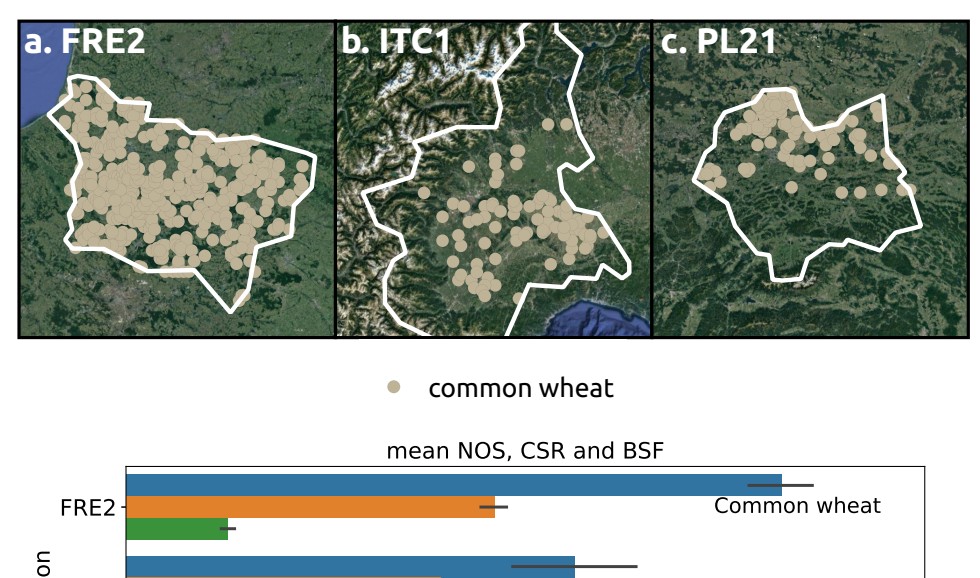

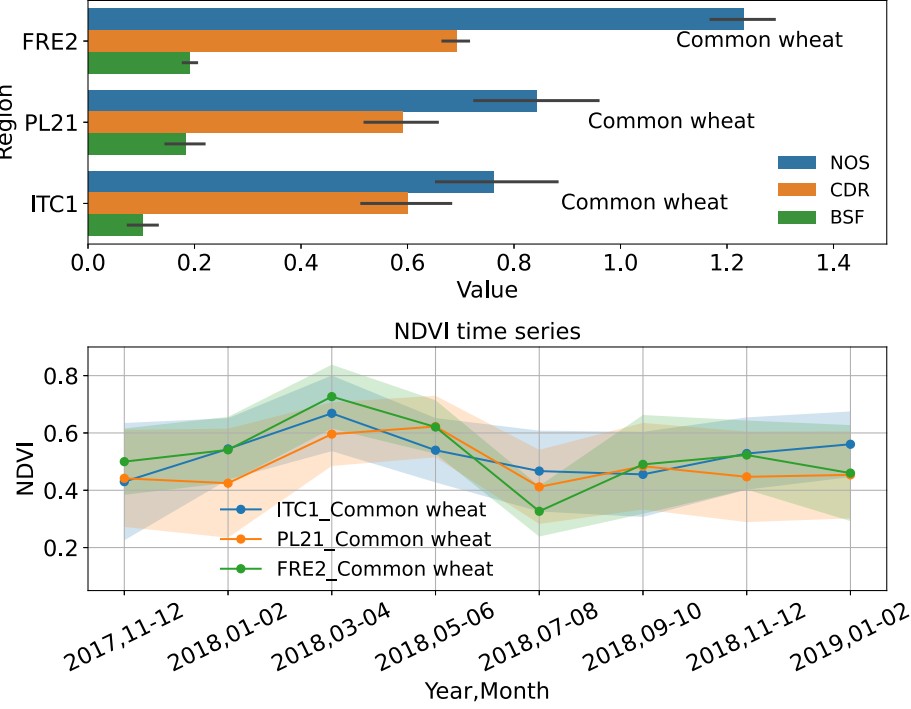

**Figure 12.** Illustration of the locations of LUCAS 2018 survey points of common wheat (top panel); statistics of various indices calculated from these LUCAS 2018 points of common wheat across regions (middle panel); and average NDVI dynamics for the year 2018, including adjacent values from neighbouring years (bottom panel). The LUCAS 2018 points data are from the same 3 NUTS2 regions as in Fig. 11. Satellite imagery source: © Google Earth (https://earth.google.com/web/).

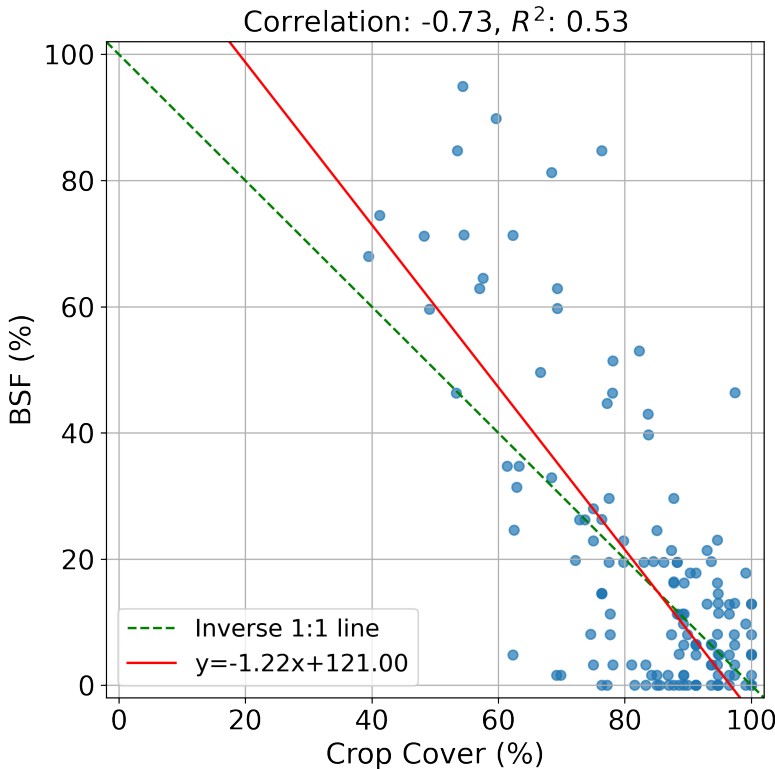

**Figure 13.** Relationship between crop cover proportion and BSF from 2007 to 2016, depicted with an inverse correlation measured by Pearson's product moment correlation coefficient and coefficient of determination $R^2$.

### 3.7 LC classification results compared to EcoDataCube.eu

Both LC models achieved a weighted F1–score of 0.77, indicating similar performance. However, the model trained on this Landsat-based predictor data cube (*i.e.* 6 bimonthly and 3 annual Landsat features per band) achieved a macro F1–score of 0.47, versus a macro F1-score of 0.45 for the model trained on Landsat bands from EcoDataCube.eu (*i.e.* 4 quarterly Landsat data with 3 percentiles per quarter). This is mainly due to the higher classification accuracy for the relatively rare *"wetlands"* class (202 points, see Fig. 17).

### 3.8 Feature importance analysis of SOC regression and LC classification

#### 3.8.1 Examination across tiers

From Table 5, we see the performance of the SOC regression model evaluated using different combinations of predictors categorized by tiers. When models are trained solely on one tier of predictors, tier 4 predictors (long-term characteristics) outperform the other tiers across all three metrics. In contrast, tier 2 predictors (bimonthly spectral indices) show the least

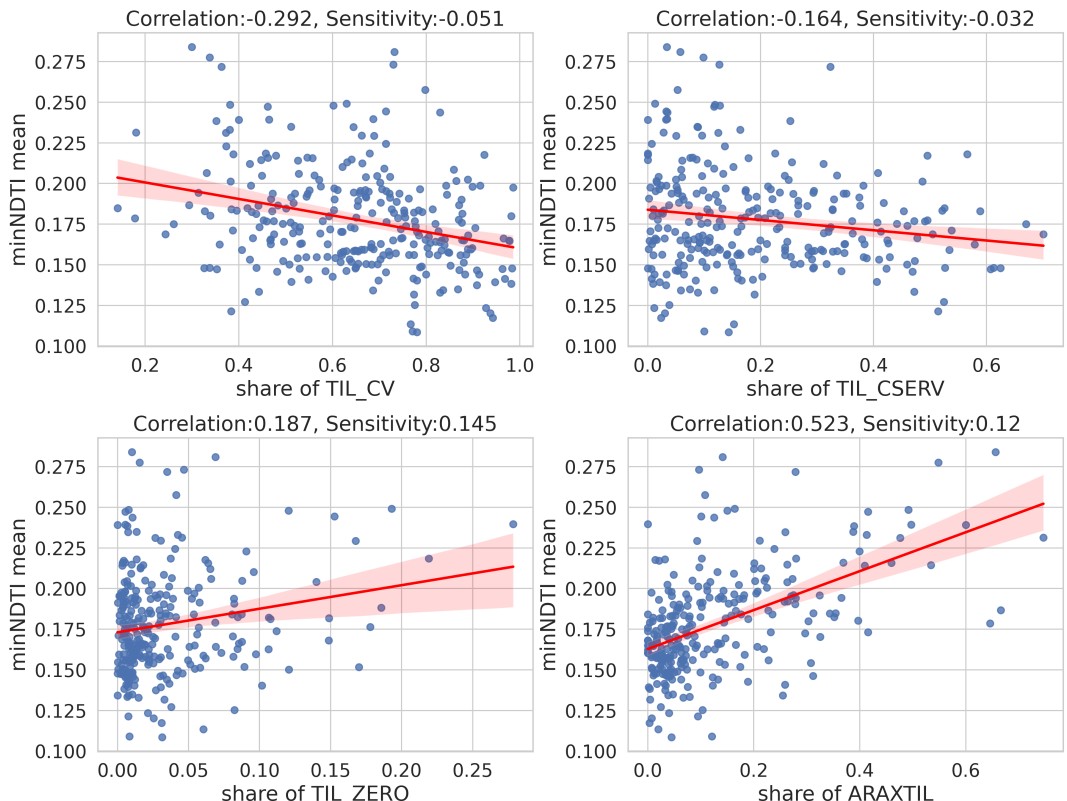

**Figure 14.** Relationship between individual share of tillage practices and average minNDTI of NUTS2 regions recorded in Eurostat. Each point represents a NUTS2 region. Four types of tillage practices shown are TIL_CV: conventional tillage; TIL_CSERV: conservation tillage; TIL_ZERO: zero tillage; ARAXTIL: land excluding from tillage survey. Coefficient refers to the Pearson correlation coefficient, quantifying the linear correlation between two variables. Sensitivity measures how changes in tillage practices influence minNDTI, obtained as a regression slope coefficient from the OLS model fitting.

optimal CCC and R-square, while tier 1 predictors (bimonthly reflectance bands) have the least optimal RMSE. For models trained on combinations of two tiers of predictors, the combination of tier 1 and tier 4 predictors performs best across all three metrics, while the combination of tier 2 and tier 4 predictors performs the least well. When examining combinations of three tiers of predictors, most metrics are quite close, but the combination of tier 1, tier 2, and tier 4 still stands out slightly. When

5   comparing across the number of tier sets combined, the general observation is that the more predictors combined, the better the model performed, with one exception: the model trained only on tier 4 predictors, despite being the smallest feature set, surprisingly achieved the best performance across all combinations.

An examination of the model performance across different tiers, as shown in Fig. 18, reveals that including tier 1 and tier 2 features generally has a negative effect. In contrast, incorporating tier 3 and tier 4 features positively impacts performance.

10   Notably, tier 4 features show to have the most positive impact on model performance, consistent with the findings from Table 5.

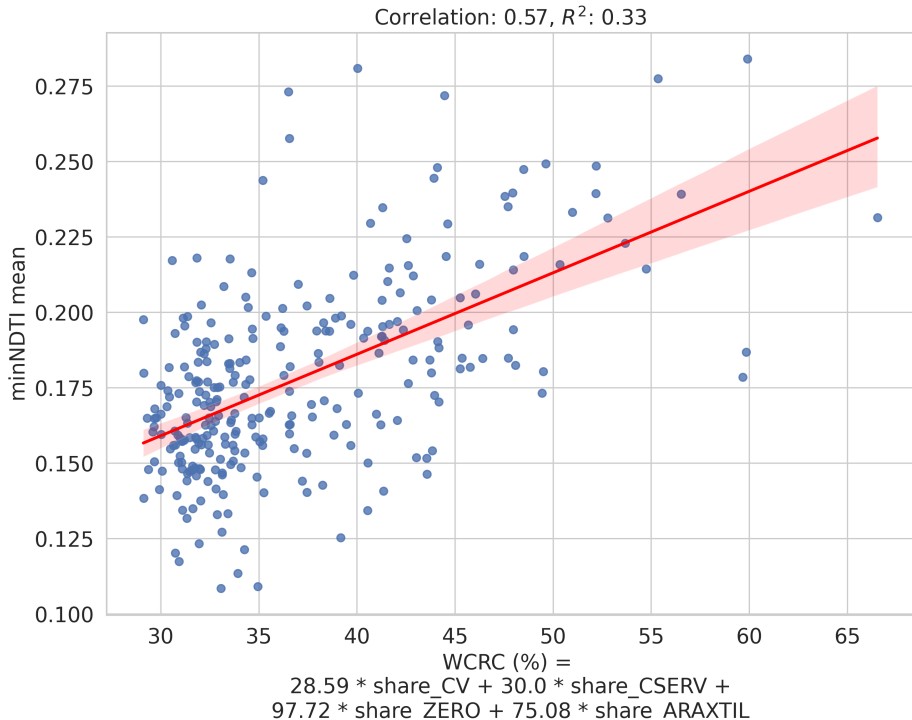

**Figure 15.** Relationship between weighted crop residue cover (WCRC) and mean minNDTI values of NUTS2 regions recorded in Eurostat. Each point stand for a NUTS2 region. Below the figure, the equation for calculating WCRC is presented, where the coefficients correspond to the fitted CRC for each type of tillage practice. Four types of tillage practices considered are TIL_CV: conventional tillage; TIL_CSERV: conservation tillage; TIL_ZERO: zero tillage; ARAXTIL: land excluding from tillage survey.

Unlike the SOC regression experiment, the LC classification experiment indicates that combining features from tier 1, tier 2, and tier 4 results in the highest accuracy (see Table 6). Although tier 4 is the best single-tier feature set for models in terms of F1-score, precision, and recall, it does not outperform all possible feature combinations. Models utilizing tier 3 predictors (annual spectral bands and indices) show the least optimal performance. For models using two-tier combinations, the pairing of tier 1 and tier 4 yields the best results across all three metrics, followed by the combination of tier 2 and tier 4. When examining models that incorporate three tiers, the combination of tier 1, tier 2, and tier 4 stands out, achieving the highest F1-score, precision, and recall among all possible feature set combinations, even slightly outperforming the feature set with all features.

Fig 19 shows that including tier 1 (bimonthly reflectance bands), tier 2 (bimonthly indices), and tier 4 (long-term characteristics) all improve model performance. Similar to the SOC regression results, models incorporating long-term trends (tier 4) exhibit the best performance metrics on average, while models including annual features tend to decrease model performance.

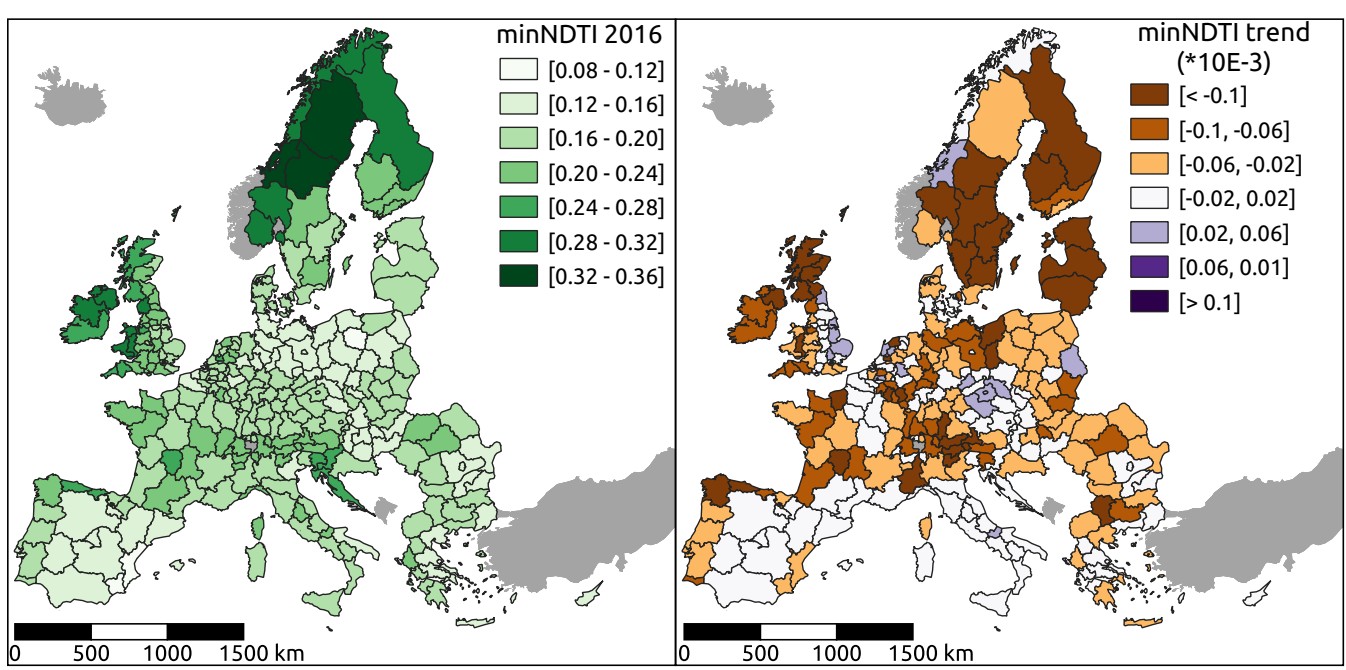

**Figure 16.** Average minNDTI values in the year 2016 (left panel) and average minNDTI trend values (right panel) by NUTS2 regions.

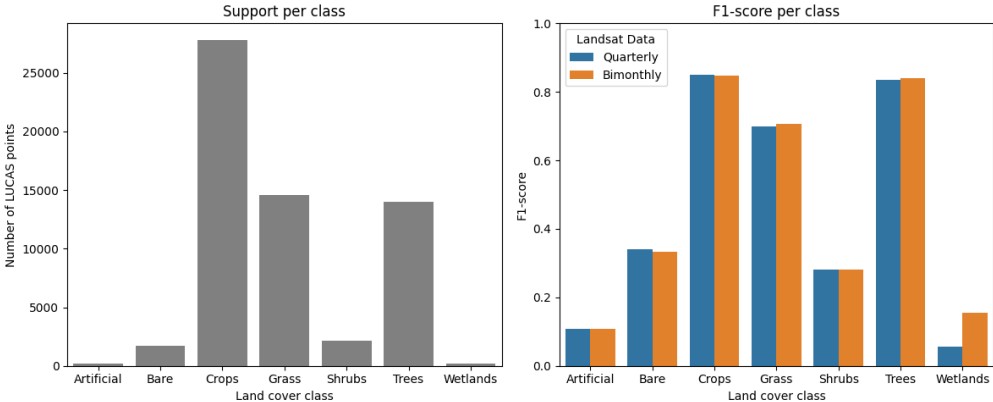

**Figure 17.** Comparison of classification results between Landsat data aggregated to 4 quarters and 3 percentiles per quarter (EcoDataCube Witjes et al. (2023)) and the data from this work, aggregated to 6 bimonthly variables per band, as well as 3 annual percentiles. Note that overall accuracy was similar, except for the 'Wetlands' class, which was classified more accurately by the model trained on the bimonthly and annual data.

**Table 5.** Metrics for SOC regression using different combinations of predictor based on tiers.

| Number of tiers combined | Tiers | Feature number | CCC | RMSE | R2 |
|---|---|---|---|---|---|
| 1 | tier1 | 42 | 0.635 | 0.651 | 0.489 |
| | tier2 | 36 | 0.623 | 0.660 | 0.476 |
| | tier3 | 38 | 0.637 | 0.652 | 0.487 |
| | tier4 | 13 | **0.682** | **0.613** | **0.547** |
| 2 | tier1+tier2 | 78 | 0.647 | 0.642 | 0.503 |
| | tier1+tier3 | 80 | 0.648 | 0.642 | 0.503 |
| | tier1+tier4 | 55 | 0.661 | 0.630 | 0.521 |
| | tier2+tier3 | 74 | 0.648 | 0.643 | 0.501 |
| | tier2+tier4 | 49 | 0.644 | 0.643 | 0.502 |
| | tier3+tier4 | 51 | 0.659 | 0.634 | 0.516 |
| 3 | tier1+tier2+tier3 | 116 | 0.652 | 0.639 | 0.507 |
| | tier1+tier2+tier4 | 91 | 0.664 | 0.629 | 0.523 |
| | tier1+tier3+tier4 | 93 | 0.663 | 0.630 | 0.522 |
| | tier2+tier3+tier4 | 87 | 0.662 | 0.631 | 0.520 |
| 4 (all predictors) | tier1+tier2+tier3+tier4 | 129 | 0.665 | 0.629 | 0.523 |

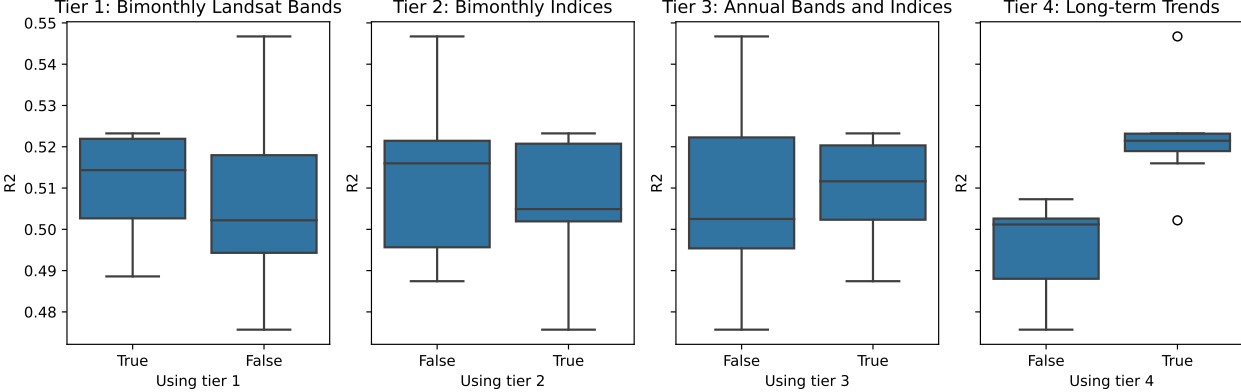

**Figure 18.** Accuracy comparison of SOC regression models trained on 15 combinations of 4 tiers of predictors.

### 3.8.2 Examination across themes

Table 7 presents the performance of the model for different combinations of predictors based on themes. When examining models trained solely on one predictor theme, reflectance band predictors perform the best across all metrics, while crop

**Table 6.** Metrics for LC classification using different combinations of predictors based on tiers.

| Number of tiers combined | Tiers | Feature number | F1-score | Precision (U.A.) | Recall (P.A.) |
|---|---|---|---|---|---|
| 1 | tier1 | 42 | 0.769 | 0.776 | 0.784 |
| | tier2 | 36 | 0.776 | 0.781 | 0.789 |
| | tier3 | 38 | 0.761 | 0.762 | 0.773 |
| | tier4 | 13 | 0.785 | 0.782 | 0.791 |
| 2 | tier1+tier2 | 78 | 0.782 | 0.787 | 0.794 |
| | tier1+tier3 | 80 | 0.774 | 0.777 | 0.787 |
| | tier1+tier4 | 55 | 0.785 | 0.792 | 0.799 |
| | tier2+tier3 | 74 | 0.777 | 0.781 | 0.790 |
| | tier2+tier4 | 49 | 0.785 | 0.791 | 0.797 |
| | tier3+tier4 | 51 | 0.775 | 0.777 | 0.787 |
| 3 | tier1+tier2+tier3 | 116 | 0.779 | 0.783 | 0.792 |
| | tier1+tier2+tier4 | 91 | **0.790** | **0.796** | **0.802** |
| | tier1+tier3+tier4 | 93 | 0.782 | 0.785 | 0.795 |
| | tier2+tier3+tier4 | 87 | 0.784 | 0.788 | 0.796 |
| 4 (all predictors) | tier1+tier2+tier3+tier4 | 129 | 0.786 | 0.789 | 0.798 |

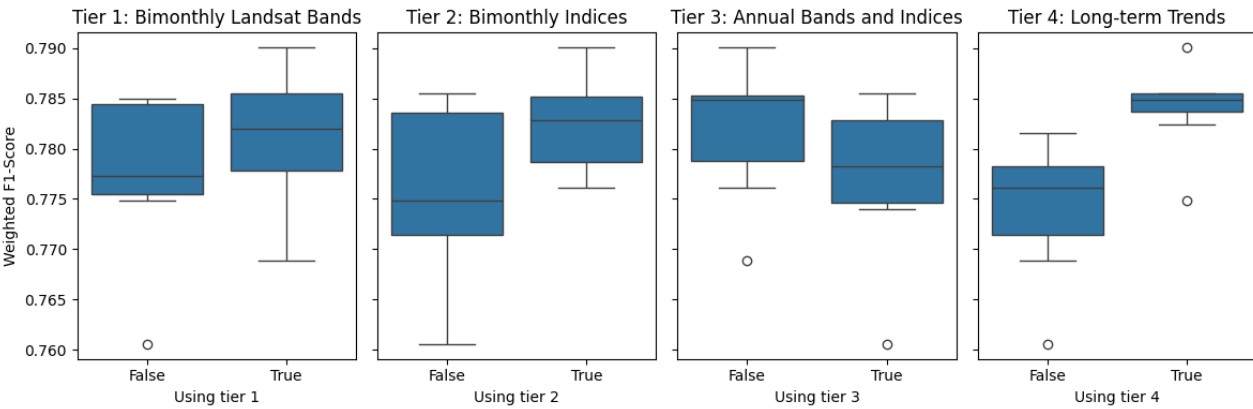

**Figure 19.** Accuracy comparison of LC classification models trained on 15 combinations of 4 tiers of predictors.

predictors perform the worst. It is important to note that crop predictors only include two variables: NOS and CDR. For models trained with combinations from two themes, those involving band predictors generally perform much better than others, especially when combined with water and vegetation predictors. In contrast, the inclusion of crop predictors appears to degrade the performance of the model. When three predictor themes are combined, the best performance is found with the combination

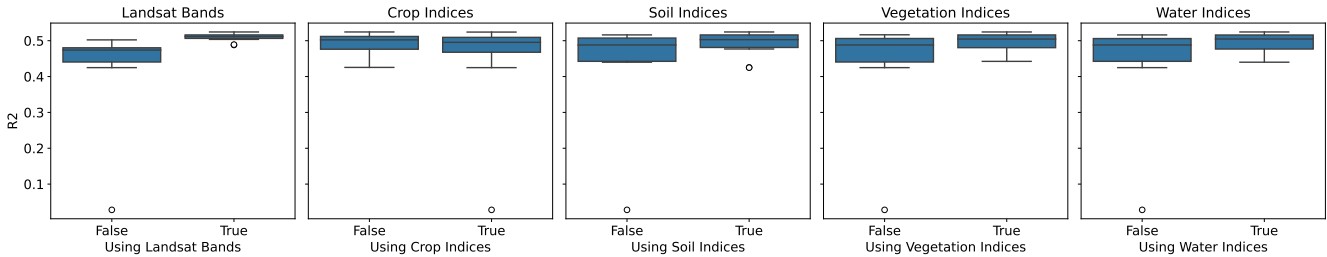

**Figure 20.** Accuracy comparison of SOC regression models trained on 31 combinations of 5 themes of predictors.

of band, soil, and water predictors, closely followed by combinations of band, soil, and vegetation, as well as band, water, and vegetation. Models lacking band predictors but including crop predictors perform worse, which is also true for models trained on four themes. Adding crop predictors to models that already include all other predictors does not significantly change the performance of the model. When comparing model performance across all possible theme-based combinations, it is evident that crop predictors do not add value to the model and can even harm performance when insufficient other predictors are available. In contrast, soil, vegetation, and water predictors all enhance the model, and the combination of these three themes provides the most significant improvement in performance.

Fig. 20 shows the R-square comparison of using or not using features from a certain theme as input for the SOC regression model. Including more thematic groups of features almost always bring in added value, especially reflectance bands, except for crop indices. The general performance patterns of the LC classification models, trained on different feature sets, resemble those of the SOC regression models (see Table 8). For LC classification models trained on a single theme feature set, crop theme features result in the worst metrics, while reflectance bands yield the best performance. Similarly to SOC regression, two-theme combinations that include band theme features generally perform better. For three-theme combinations, the combination of band, soil, and water themes is the most effective. For models trained on combinations of four themes, the best performance is observed when crop features are excluded. In particular, the model trained on the entire feature space achieved the highest accuracy metrics, with a minimal difference to the model trained on all features except crop indices, specifically using the combination of band, soil, vegetation and water themes (0.786331 versus 0.786330).

On average, models that incorporate Landsat bands achieved the highest F1-scores, while those including crop indices had the lowest, as shown in Fig. 21. Including additional thematic predictors, in addition to crop indices, generally improved model performance.

### 3.8.3 Comparison of feature space between SOC regression and LC classification

Comparing the results of the tier-based and theme-based for both SOC regression and LC classification, the improvement in model performance from combining more tiers is less significant than the enhancement seen when increasing the number of combined themes. The contribution of features from different tiers varies between SOC regression and LC classification: including tier 3 and tier 4 features improves the SOC regression model, while tier 3 is the only feature tier that decreases model

**Table 7.** Metrics for SOC regression using different combinations of predictors based on themes.

| Number of themes combined | Feature combination | Feature number | CCC | RMSE | R2 |
|---|---|---|---|---|---|
| 1 | band | 63 | 0.635 | 0.652 | 0.488 |
| | crop | 2 | 0.061 | 0.898 | 0.028 |
| | soil | 17 | 0.582 | 0.690 | 0.426 |
| | vegetation | 27 | 0.594 | 0.680 | 0.443 |
| | water | 20 | 0.591 | 0.681 | 0.441 |
| 2 | band+crop | 65 | 0.636 | 0.651 | 0.489 |
| | band+soil | 80 | 0.649 | 0.642 | 0.504 |
| | band+vegetation | 90 | 0.651 | 0.639 | 0.508 |
| | band+water | 83 | 0.651 | 0.639 | 0.508 |
| | crop+soil | 19 | 0.581 | 0.691 | 0.425 |
| | crop+vegetation | 29 | 0.593 | 0.680 | 0.442 |
| | crop+water | 22 | 0.590 | 0.682 | 0.440 |
| | soil+vegetation | 44 | 0.631 | 0.655 | 0.483 |
| | soil+water | 37 | 0.626 | 0.658 | 0.479 |
| | vegetation+water | 47 | 0.618 | 0.661 | 0.474 |
| 3 | band+crop+soil | 82 | 0.650 | 0.641 | 0.505 |
| | band+crop+vegetation | 92 | 0.650 | 0.640 | 0.507 |
| | band+crop+water | 85 | 0.651 | 0.639 | 0.507 |
| | band+soil+vegetation | 107 | 0.659 | 0.634 | 0.516 |
| | band+soil+water | 100 | 0.660 | 0.634 | 0.516 |
| | band+vegetation+water | 110 | 0.658 | 0.634 | 0.516 |
| | crop+soil+vegetation | 46 | 0.630 | 0.656 | 0.482 |
| | crop+soil+water | 39 | 0.625 | 0.659 | 0.477 |
| | crop+vegetation+water | 49 | 0.621 | 0.659 | 0.476 |
| | soil+vegetation+water | 64 | 0.646 | 0.643 | 0.502 |
| 4 | band+crop+soil+vegetation | 109 | 0.660 | 0.634 | 0.516 |
| | band+crop+soil+water | 102 | 0.660 | 0.633 | 0.517 |
| | band+crop+vegetation+water | 112 | 0.657 | 0.634 | 0.516 |
| | band+soil+vegetation+water | 127 | **0.666** | **0.628** | **0.524** |
| | crop+soil+vegetation+water | 66 | 0.645 | 0.643 | 0.502 |
| 5 (all predictors) | band+crop+soil+vegetation+water | 129 | **0.666** | **0.628** | **0.524** |

**Table 8.** Metrics for LC classification using different combinations of predictors based on themes.

| Number of themes combined | Themes | Features | F1-score | Precision | Recall |
|---|---|---|---|---|---|
| 1 | band | 63 | 0.769 | 0.774 | 0.784 |
| | crop | 2 | 0.288 | 0.308 | 0.458 |
| | soil | 17 | 0.683 | 0.685 | 0.700 |
| | vegetation | 27 | 0.741 | 0.744 | 0.756 |
| | water | 20 | 0.724 | 0.727 | 0.739 |
| 2 | band+crop | 65 | 0.771 | 0.776 | 0.785 |
| | band+soil | 80 | 0.778 | 0.783 | 0.792 |
| | band+vegetation | 90 | 0.775 | 0.780 | 0.789 |
| | band+water | 83 | 0.777 | 0.781 | 0.790 |
| | crop+soil | 19 | 0.683 | 0.685 | 0.700 |
| | crop+vegetation | 29 | 0.741 | 0.745 | 0.756 |
| | crop+water | 22 | 0.725 | 0.730 | 0.741 |
| | soil+vegetation | 44 | 0.762 | 0.764 | 0.775 |
| | soil+water | 37 | 0.749 | 0.753 | 0.762 |
| | vegetation+water | 47 | 0.769 | 0.773 | 0.781 |
| 3 | band+crop+soil | 82 | 0.778 | 0.784 | 0.792 |
| | band+crop+vegetation | 92 | 0.774 | 0.780 | 0.788 |
| | band+crop+water | 85 | 0.776 | 0.781 | 0.789 |
| | band+soil+vegetation | 107 | 0.781 | 0.785 | 0.794 |
| | band+soil+water | 100 | 0.784 | 0.788 | 0.797 |
| | band+vegetation+water | 110 | 0.780 | 0.785 | 0.793 |
| | crop+soil+vegetation | 46 | 0.760 | 0.763 | 0.773 |
| | crop+soil+water | 39 | 0.747 | 0.751 | 0.761 |
| | crop+vegetation+water | 49 | 0.769 | 0.774 | 0.782 |
| | soil+vegetation+water | 64 | 0.779 | 0.782 | 0.791 |
| 4 | band+crop+soil+vegetation | 109 | 0.781 | 0.785 | 0.794 |
| | band+crop+soil+water | 102 | 0.784 | 0.789 | 0.797 |
| | band+crop+vegetation+water | 112 | 0.780 | 0.785 | 0.793 |
| | band+soil+vegetation+water | 127 | **0.786** | **0.790** | **0.799** |
| | crop+soil+vegetation+water | 66 | 0.778 | 0.782 | 0.791 |
| 5 (all predictors) | band+crop+soil+vegetation+water | 129 | **0.786** | **0.790** | **0.799** |

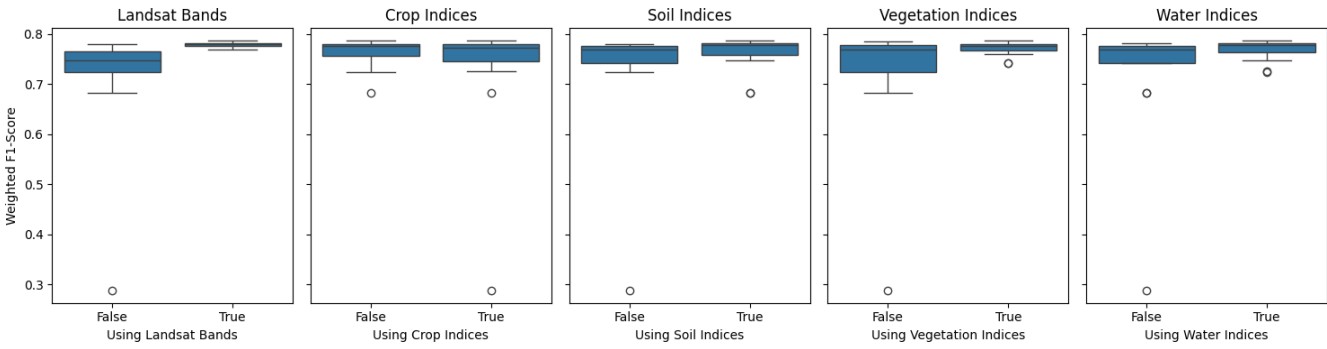

**Figure 21.** Accuracy comparison of LC classification models trained on 31 combinations of 5 themes of predictors.

performance for LC classification. In both experiments, tier 4 demonstrates its importance. Regarding the feature examination across themes, both experiments show similar results: reflectance bands provide a solid foundation for the models, and different thematic features add valuable information. However, crop indices contribute very little and can even degrade model performance when other variables are insufficient.

## 4 Discussion

### 4.1 Comparison to other open EO datasets

Our ARCO data cube, with almost 17TB of data for Europe at a 30 m spatial resolution from 2000 to 2022, represents a significant improvement over existing pan-EU EO data sets in terms of completeness, consistency, and comprehensiveness. Through an automated workflow, we ensured that all data layers adhere to uniform standards in naming conventions, projection systems, spatial extents, and resolutions. This consistency allows users to easily combine layers over space and time, enabling detailed studies using various indices. Our multidimensional index data cube, optimized as ARCO mosaics to be cloud-free and gap-free, significantly reduces preprocessing efforts compared to using raw data from open data archives such as Sentinel (https://dataspace.copernicus.eu/explore-data/data-collections/sentinel-data), MODIS (https://modis.gsfc.nasa.gov/data/dataprod/), or Landsat (https://www.usgs.gov/products/data), which are freely provided by agencies like USGS, NASA, and the European Commission's Copernicus and Sentinel programs.

The concept of ARD has been applied in projects to manage multidimensional data cubes from remote sensing images for specific countries and regions, such as the Australian Data Cube (Lewis et al., 2017), Swiss Data Cube (Giuliani et al., 2017), and Africa Regional Data Cube (Killough, 2019). Although Witjes et al. (2023) offers the first pan-European data cube, it is limited to quarterly temporal resolution and only cloud-optimized reflectance bands. Our data cube extends this work by refining the temporal resolution and developing four tiers of predictors. In addition to bimonthly indices that capture detailed environmental dynamics, annual aggregations enable yearly comparisons, and long-term characteristics represent general states and trends of vegetation, water, and crop management across Europe. This advancement benefits from two advantages of the

Landsat program, the world's longest continuously running EO mission and detailed 30 m spatial resolution (Wulder et al., 2022). These indices complement ground survey data, which often lack temporal continuity and are limited in duration due to the significant time and effort required.

Although some data sets provide continuous time series of spectral indices, they are often scattered and inconsistent in format, resolution, and coverage (Wagemann et al., 2021). For instance, most reconstruction efforts focus on NDVI (Zhou et al., 2016), while indices such as NOS, CDR, and BSF are less available (Estel et al., 2015, 2016; Ettehadi Osgouei et al., 2019; Demattê et al., 2020). Our data cube's multidimensional nature provides comprehensive insights into various aspects. The BSF index, for example, helps in crop management by mitigating soil exposure risks and revealing larger land use changes such as deforestation (Demattê et al., 2020). Combined indices provide a holistic view, *e.g.* with NOS and CDR, along with BSF, detailing crop cycles and identifying intensively cultivated areas (Estel et al., 2016). Coupled with climate and human footprint data, these indices facilitate studies of the interactions between climate, human activities, and environments (Brown et al., 2012; Seifert and Lobell, 2015).

Although our Landsat indices data cube encompasses a wide range of indices, it is less specialized than studies designed to quantify specific processes. For example, BSF, proposed in previous work (Demattê et al., 2020; Mzid et al., 2021), is mapped here for the first time annually across continental Europe as a complete, gap-free time series and has proven useful in modeling experiments. However, it is not a direct representation of soil surface. In contrast, many studies mapping bare soil focus on identifying the "barest" pixel to create a Bare Soil Composite (BSC) (Diek et al., 2017; Rogge et al., 2018; Safanelli et al., 2020; Demattê et al., 2020; Mzid et al., 2021; Rizzo et al., 2023). These studies explored more sophisticated methods to ensure the algorithm accurately identifies truly bare pixels while effectively filtering out non-soil signals, such as vegetation. Typically, this involves a combination of indices and, in some cases, land use masks (Rogge et al., 2018; Diek et al., 2017; Safanelli et al., 2020; Heiden et al., 2022). As a result, these BSC products provide a more direct and reliable representation of soil properties. While its applicability may be limited in regions dominated by permanent vegetation (Hill and Guerschman, 2022) and it is typically produced as a static snapshot, BSC remains a valuable predictor for digital soil mapping (DSM) (Broeg et al., 2024). Recent studies suggest that combining the strengths of BSF and BSC is feasible. For instance, Zepp et al. (2023) investigated the temporal windows required to generate recurrent BSC products, highlighting the potential to create BSC time series that capture dynamic soil surface properties. Building on this, a natural next step for this study is to apply this approach and generate BSC time series as a new feature, leveraging the extensive Landsat archive at our disposal. Additionally, BSF could be integrated as an indicator to dynamically refine the temporal frame for BSC generation.

Many studies on Land Surface Phenology (LSP), *i.e.*, the descriptors of spatio-temporal patterns of the vegetated land surface derived from EO data (De Beurs and Henebry, 2005), present different methods to quantify crop cycles (Henebry and De Beurs, 2013; Estel et al., 2016; Liu et al., 2020; Meroni et al., 2021). All these methods highlight potential avenues for refining the current CDR and NOS data products in the future. Most LSP studies and products have moderate to coarse spatial resolutions (Ganguly et al., 2010; Estel et al., 2016; Zhang et al., 2018; Caparros-Santiago et al., 2021), due to the trade-off between spatial resolution and revisit frequency (Bolton et al., 2020; Meroni et al., 2021; Caparros-Santiago et al., 2021). Although the Harmonized Landsat Sentinel (HLS) product—created by harmonizing data from Landsat-8/9 and Sentinel-2A/B—offers

new opportunities to map LSP indicators at fine spatial resolutions (Bolton et al., 2020), it is only available starting from 2013. Lastly, despite the potential to quantify the intensity of tillage with EO data such as Landsat-7/8/9 and Sentinel-2, the availability of information on tillage practice on a continental scale remains limited.

In summary, the Landsat indices we described in this paper can be seen as part of a comprehensive effort to produce useful biophysical indices to support soil, land, water and land use monitoring. While our methods may be less specialized than those of dedicated thematic studies, this reflects a compromise to balance computational resource demands and the need for complete, gap-free coverage. Despite this, they are designed as a foundational resource, offering a broad and adaptable set of predictors in a comprehensive, gap-free form that requires minimal effort from users for effective application in various environmental modeling and monitoring tasks.

## 4.2   Implications from plausibility checks

Our quality assessment primarily focused on third and fourth-tier predictors, such as minNDTI, BSF, NOS, CDR and their long-term trends, due to their complexity and the availability of relevant ground data. We also visualized the annual P50 of NDVI and NDWI to provide basic information on general vegetation and water conditions. Visual examination indicated that annual NDVI, NDWI, and BSF effectively differentiate various landscapes, while NOS and CDR distinguish different crop patches. Although interpreting these indices without detailed crop management information is challenging, regional statistics of NOS, CDR, and BSF reveal significant differences in crop cycles among different crop types and regions across Europe. We demonstrated that the BSF index effectively measures cropland bareness, with a high negative correlation coefficient of -0.73 with the duration of crop cover in agricultural areas between 2007 and 2016. However, the moderate correlation of minNDTI with tillage practices statistics from EUROSTAT suggests the need for further investigation.

In Fig. 5, a negative NDVI trend is observed in regions characterized by high altitude and latitude. This finding appears to be in conflict with the greening phenomenon in cold / snowy ecosystems, as reported by Obuchowicz et al. (2023). Furthermore, the positive trend observed in the NDWI trend map is inconsistent with Poussin et al. (2021)'s observation of a slight decline in NDWI values in the Alps region of Switzerland during the period from 1984 to 2019. Several factors could be influencing these discrepancies, such as the different temporal scales used for trend calculation, as noted by de Jong and de Bruin (2012), or the increased dependence on gap-filling due to limited available high-quality EO data in these regions. These contrasting observations call for further investigation to understand the underlying causes and implications of them.

Although these efforts are limited by the availability of relevant ground data and sometimes yield inconclusive results, it is important to note that these indices are proxies for ground objects and processes, rather than exact measurements of ground conditions. They are designed to complement, not replace, ground survey data. Therefore, plausibility checks are not expected to provide one-to-one validation but to highlight the advantages and limitations of these indices. In summary, well-established indices such as NDVI and NDWI, and the relatively straightforward BSF index, are reliable and suitable for direct analysis, as demonstrated in Sect. 3.3. However, for more complex indices like NOS and CDR, robust assessments require careful integration with ground-truth data to ensure accuracy and validity.

### 4.3 Insights from modelling experiments

By looking at predictors of different themes' impact on SOC regression and LC classification model, band theme predictors are vital for both models. This is unsurprising, as they form the foundation from which other spectral indices are derived and provide a general representation of surface objects and processes (Zhou et al., 2021). Apart from band theme predictors, models trained on single-theme predictors show that vegetation predictors have a stronger positive impact on both LC and SOC modeling than other themes. Vegetation is fundamental in defining land covers, naturally ranking high in feature importance. Its close relationship with living soil, through through physical, chemical, and biological interactions (Kooch et al., 2022), positions it as a vital *"organism"* factor in the soil formation process and a key driver of SOC change (McBratney et al., 2003; Wiesmeier et al., 2019). This importance is reflected in numerous SOC modeling studies, where vegetation indices like NDVI and EVI consistently rank among the top predictors, sometimes even used as the only spectral index (Gomes et al., 2019; Sothe et al., 2022; Zeraatpisheh et al., 2022; Wadoux et al., 2020).

Soil and water theme parameters also contribute valuable insights for SOC modeling. The indices of these two themes already differ statistically across most LCs, even before considering LC classification (Szabo et al., 2016). For SOC modeling, tillage practices and soil exposure, as indicated by soil theme predictors, negatively alter soil structure and impact carbon dynamics (Chahal et al., 2021; Yang et al., 2020b; Demattê et al., 2020; Mzid et al., 2021; Wiesmeier et al., 2019). Similarly, water dynamics and climate, represented by water theme predictors, are key elements of the *climate"* factor in the soil-forming *"scorpan"* model (Maynard and Levi, 2017; Wiesmeier et al., 2019; Zeraatpisheh et al., 2022). When direct climate data, such as precipitation and temperature, are unavailable, water theme indices can serve as useful substitutes.

Combining predictors from multiple themes generally enhances model performance. This is consistent with the findings of Witjes et al. (2023) for LC classification. For SOC modeling, most studies in DSM ensure that all soil-forming factors are incorporated within the *"scorpan"* model (McBratney et al., 2003; Wiesmeier et al., 2019; Wadoux et al., 2020). Zeraatpisheh et al. (2022) demonstrated that while combining feature groups may not always result in higher prediction accuracy, it does produce SOC predictions with lower uncertainty. The exception is crop theme predictors, likely due to limitations in our methods for quantifying NOS and CDR. Numerous studies have shown that phenological parameters can improve soil property and LC modeling, at least for croplands (Maynard and Levi, 2017; Yang et al., 2020a; He et al., 2021; Xue et al., 2014; Nguyen et al., 2020). However, these studies often focus on small areas with much less data, allowing for localized and detailed extraction of phenological parameters. In contrast, our approach to extracting phenological parameters like NOS and CDR, and assessing their relevance to SOC and LC modeling, remains limited. Further investigation is needed to determine whether phenological parameters can be effectively scaled to large, continental-level studies and, if so, to determine the methods for their application.

Unlike theme-based predictors, predictors of different tiers exhibit varying levels of importance for SOC regression and LC classification modeling. For SOC regression, predictors that represent longer periods are more important, aligning with the findings of Zepp et al. (2023). This is clear in Fig. 18, where predictors with extended temporal processing significantly enhance model performance. Table 5 shows that model trained solely on tier 4 predictors (long-term characteristics) consistently

outperform others across all three metrics. This suggests that long-term characteristics are highly valuable for SOC regression and can sometimes outperform more complex combinations of features. A study by Yang et al. (2022) demonstrates that temporal variables are effective in modeling temporal SOC variation, though their scope spans 30 years, while the LUCAS topsoil survey covers only nine years. Dynamic information appears less beneficial for SOC modeling, which is not surprising since SOC is a slow-changing soil property that typically varies over decades.

In contrast, incorporating detailed dynamic information (tier 1 and tier 2: bimonthly bands and indices) improves the model performance for LC classification. This improvement may be attributed to the dynamic nature of LC, which changes more frequently or suddenly compared to SOC, making it better captured with more frequent revisit information. The temporal dimension has always been critical in LC classification. Researchers have emphasized incorporating temporal features into models to enable the mapping of LC at finer temporal scales, such as annual intervals (Gómez et al., 2016; Zhang et al., 2020; Brown et al., 2022). Surprisingly, the tier 3 (annual) indices are shown to be less important for LC classification. This could be because the temporal characteristics within a year neither provide the detailed phenological information that bimonthly predictors do (as phenological process aren't defined strictly by a calender year), nor offer the long-term characteristics that tier 4 predictors provide.

Combining predictors from multiple themes adds more value than combining predictors from multiple tiers. This is likely because different tiers mainly vary in their level of temporal feature engineering, conveying similar messages. However, more predictors do not always lead to better performance. However, as demonstrated by our SOC regression and LC classification experiments, using more features does not always lead to better performance. This can increase the risk of multicollinearity and overfittin, as well as add complexity and require greater computational resources (Wadoux et al., 2020). Feature importance analysis highlights that feature selection should be guided by a clear understanding of the predictors' characteristics, modeling objectives, and task constraints. Factors such as the temporal and spatial resolution of covariates, the spatial extent of modeling, and the balance of sample data also affect the determination of the appropriate features (Chen et al., 2022; Shi et al., 2018; Nazari et al., 2020).

## 4.4 Limitations

As with any product derived from the Landsat ARD V2, users should be aware of inherent limitations. A key step in harmonizing Landsat products into ARD is surface reflectance normalization (Potapov et al., 2020), which differs from physically-based atmospheric and surface anisotropy corrections. Our plausibility checks demonstrate that the bimonthly Landsat reflectance bands (tier-1) are well-suited for index derivation and modeling purposes, indicating sufficient product quality. However, similar to the Landsat ARD V2 data, this data cube may not be suitable for very precise studies of land surface reflectance.

As noted by Potapov et al. (2020), the GLAD Landsat ARD algorithm struggles with wintertime image processing in temperate and boreal climates at high latitudes. This is due to low sun azimuth angles and the similarity between snow cover and clouds during the winter season. As a result, some images and their resulting 16-day composites may still contain snow-covered observations. To address this issue, additional filtering—such as applying the "snow/ice" observation quality flag—was implemented in this study to remove affected observations. However, this process resulted in larger data gaps in certain regions.

Consequently, the TSIRF algorithm, which depends on other images in the sequence for reconstruction, can produce inaccurate results when the time series contains too few valid pixels (Consoli et al., 2024). This is also reflected in the reduced time series reconstruction accuracy of the European dataset compared to the global dataset, which can be attributed to Europe's higher average latitude compared to the global dataset. Consequently, artifacts are observed in high-altitude and high-latitude regions, such as central Sweden and southern Norway, as illustrated in Figures 5 and 6. While these artifacts do not significantly impact the overall analysis, users should be aware of them and exercise caution when interpreting results, especially in regions with challenging data conditions.

As with many Land Surface Phenology (LSP) studies, our temporally-aggregated indices such as BSF, NOS, and CDR rely on threshold-based methods (De Beurs and Henebry, 2010). These thresholds can vary depending on crop types and regional characteristics, often requiring expert knowledge for accurate application, especially crop relevant indices (Chen et al., 2016). Given the broad spatial and temporal scope of our data cube, we adopted a generalized threshold approach, which might not capture local variations accurately (Henebry and De Beurs, 2013; Estel et al., 2016; Demattê et al., 2020). The limitation of this method lies in its inability to distinguish between meaningful changes and natural fluctuations exceeding the threshold (De Beurs and Henebry, 2010). Therefore, users are advised to proceed cautiously and conduct preliminary tests when a meaningful conclusion needs to be drawn from direct analysis with this limited local or domain knowledge.

Another challenge posed by advances in EO technology is the expanding list of satellite spectral indices, each uniquely calibrated for specific environmental attributes. For example, the Awesome Spectral Index (ASI), developed by Montero et al. (2023), lists 127 vegetation indices by its publication date. Incorporating all of these indices into a single data cube is impractical. To balance data volume and complexity, we adopted a hybrid approach. The predictors data cube includes the indices most commonly applied, covering several important aspects of the environment, such as vegetation health, soil conditions, and water dynamics. Additionally, developed in alignment with the principles of the Open Data Cube (ODC), our ARCO data cube includes guided documentation and scripts to assist users in querying, exploring, performing analyses, and tracking the provenance of all contained data. This approach facilitates quality control and updates, allowing users to customize the data cube according to their specific needs or to explore new indices that could offer additional insight into the environment.

The data cube includes a wide array of indices with various applications. However, some indices may yield irrelevant values outside their intended regions. For example, many indices lose significance in snowy environments at high latitudes and altitudes, and minNDTI does not convey useful information about tillage practices outside of arable land. It is crucial not to apply these indices indiscriminately across all land covers. Other indices within the data cube can help delineate appropriate application areas. For example, the NDSI can exclude snowy areas, ensuring a more accurate application of other indices.

## 4.5 Future work

While NOS and CDR provide useful insights into crop cycles and their duration, their limited contribution to SOC regression models suggests a need for further refinement. Improving the methods for quantifying these indices or exploring alternative indicators might enhance their utility in SOC modeling. As discussed in Section 4.1, there is significant value in learning from

specialized studies that focus on accurately quantifying specific processes. Future updates to this work will incorporate more localized and domain-specific knowledge could help to tailor these indices to better capture relevant environmental processes.

Although the original Landsat archive has a 16-day temporal revisit, we aggregated the data set into a bimonthly resolution to address quality issues such as gaps and cloudy pixels. Although bimonthly resolution effectively captures annual dynamics, a finer 16-day resolution could provide richer details, especially for NDTI values observed from harvest windows crucial for mapping tillage practices (Zheng et al., 2012). However, finer temporal resolution requires extensive gap-filling, presenting a significant trade-off between temporal detail and data consistency.

Our data set currently relies exclusively on Landsat ARD V2 to derive NOS, CDR, and minNDTI. However, integrating multiple sensors could enhance accuracy (Henebry and De Beurs, 2013; Caparros-Santiago et al., 2021). For example, using minimum surface temperature thresholds can prevent false crop cycle identifications (Li et al., 2014), and integrating precipitation data can improve crop residue estimates (Beeson et al., 2020; Dvorakova et al., 2023). Furthermore, combining Harmonized Landsat-Sentinel (HLS) data could further improve index usability, especially for years covered by Sentinel-2 (Wu et al., 2022).

As indicated by the survey results of Wagemann et al. (2021), improving the usability of spectral indices requires the participation of various stakeholders, including farmers, policymakers, and sector-specific experts. Their local knowledge and domain-specific insight can refine algorithms and improve the utility of data. This approach aligns with the objectives of the Horizon Europe AI4SoilHealth project (2023–2027), aiming to bridge the gap between EO data and practical applications.

## 5 Conclusions

In this paper, we present the production process and quality assessment of a 17TB ARCO data cube, primarily based on the Landsat indices. It is available at a 30 m resolution from 2000 to 2022, comprising multiple spectral indices. Multi-tiers of preditors are derived to represent different levels of temporal information extraction: bimonthly, annual, and long-term analyses covering continental Europe, including Ukraine, the UK, and Turkey. The main advantages of our dataset include: (1) its fine spatial resolution and extensive temporal coverage, which are crucial for analyzing and tracking the spatial variations and temporal dynamics of different environmental aspects, such as soil; (2) its broad coverage of indices, each providing unique insights into different aspects, together offering a holistic view of environment; (3) it is analysis-ready for modeling and mapping applications.

To maximize the effectiveness and reliability of the data, the following usage recommendations are proposed:

- **Established Indices**: For well-established indices like NDVI, NDWI, SAVI, FAPAR, NDSI, and those demonstrating high reliability such as BSF, the data cube provides valuable information and insights, especially when ground truth data are lacking.

- **Complex Indices**: For more complex indices that were not validated in this study due to the lack of corresponding ground truth data, caution is advised. These should be used alongside ground truth data to ensure accuracy and validity, and their application should consider domain and local knowledge.

- **Modeling**: All indices in the data cube have proven effective as predictors for modeling purposes, making them useful for various environmental and ecological models.

- **Feature Selection**: It is beneficial to perform feature selection to identify the best combinations of indices from the comprehensive predictor sets for specific modeling tasks, optimizing the model's performance and relevance to the study's objectives.

## 6  Data availability

This dataset is registered with a valid DOI (Tian et al., 2024, https://doi.org/10.5281/zenodo.10776891). While each data bucket has a unique DOI, we kindly ask users to cite the DOI on the Zenodo landing page to ensure consistency. This DOI represents all dataset versions and always resolves to the latest release.

Due to Zenodo's storage limitations and the large size of the dataset, only portions of the data are stored in Zenodo, distributed across multiple buckets. To streamline access, we have created a Zenodo landing page that includes a navigation catalog linking to the individual buckets. Additionally, a central access catalog, provided as a Google Sheet, offers complete links for public access to the data stored in Wasabi Cloud Storage. Both the Zenodo landing page and the central catalog include a detailed explanation of the naming conventions used.

To lower the barrier to data use, metadata is embedded directly into file names. Each dataset layer follows a standardized naming format, structured into 10 key fields: generic variable name, variable procedure combination, position in the probability distribution or variable type, spatial support, depth reference, time reference (including start and end times), bounding box, EPSG code, and version code. Each metadata field serves a specific purpose in assessing the datasets' fitness for use. For the datasets presented in this study, the metadata specifies a uniform spatial support of 30 m resolution, a depth reference denoted as *s* (land surface), a bounding box identified as *eu*, and an EPSG code of *EPSG:3035*. For the other fields: the generic variable name helps users identify the required predictor layer; the variable procedure combination provides indications of how the data was derived and its source; the time reference, comprising the start and end dates, ensures users can select layers matching their relevant temporal scope; and the version code, which reflects the creation date of the corresponding layer, facilitates tracking and version control. This metadata framework enhances geodata trustworthiness and usability by allowing users to assess dataset relevance and suitability directly from the name. This metadata framework enhances geodata trustworthiness and usability by allowing users to assess dataset relevance and suitability directly from the name.

## 7  Code availability

Complete code used to derive all indices and generate the visualization figures is available at https://doi.org/10.5281/zenodo.12907281. Even more spectral indices can be derived using the https://github.com/awesome-spectral-indices/ library provided by Montero et al. (2023).

## 8   Author contribution

Xuemeng Tian and Tomislav Hengl conceptualized the study. Leandro Parente, Davide Consoli, Xuemeng Tian, and Yufeng Ho curated the data. Funding was secured by Tomislav Hengl and Leandro Parente. Xuemeng Tian and Florian Schneider conducted the investigation. Methodology was developed by Xuemeng Tian, Leandro Parente, and Davide Consoli. Project administration was handled by Tomislav Hengl and Robert Minařík. Resources were provided by Tomislav Hengl and Leandro Parente. Software development was performed by Leandro Parente, Davide Consoli, Xuemeng Tian, and Yufeng Ho. Supervision was carried out by Tomislav Hengl, Leandro Parente, Florian Schneider, and Robert Minařík. Xuemeng Tian, Martijn Witjes, and Florian Schneider validated the results. Visualization was done by Murat Şahin, Xuemeng Tian, and Martijn Witjes. The original draft was prepared by Xuemeng Tian, Tomislav Hengl, Martijn Witjes and Davide Consoli, with review and editing contributions from Tomislav Hengl, Martijn Witjes, and Yufeng Ho.

## 9   Competing interests

Xuemeng Tian, Davide Consoli, Martijn Witjes, Leandro Parente, Murat Şahin, Yu-Feng Ho, Robert Minařík & Tomislav Hengl are employed by OpenGeoHub.

## 10   Disclaimer

Funded by the European Union. Views and opinions expressed are however those of the author(s) only and do not necessarily reflect those of the European Union or European Commision. Neither the European Union nor the granting authority can be held responsible for them. The data is provided "as is". AI4SoilHealth project consortium and its suppliers and licensors hereby disclaim all warranties of any kind, express or implied, including, without limitation, the warranties of merchantability, fitness for a particular purpose and non-infringement. Neither AI4SoilHealth Consortium nor its suppliers and licensors, makes any warranty that the Website will be error free or that access thereto will be continuous or uninterrupted. You understand that you download from, or otherwise obtain content or services through, the Website at your own discretion and risk.

## 11   Acknowledgements

The authors are grateful to Sytze de Bruin (Wageningen University & Research), Kirsten de Beurs (Wageningen University & Research) and Bernhard Ahrens (Max Planck Institute for Biogeochemistry) for their valuable advice on the structure, visualization, and phrasing of this manuscript. The authors are especially grateful to Calogero Schillaci (European Commission Joint Research Center) for providing constructive feedback on the first version of the manuscript.

The AI4SoilHealth project has received funding from the European Union's Horizon Europe research and innovation program under grant agreement No. 101086179. The Open-Earth-Monitor Cyberinfrastructure project has received funding from the European Union's Horizon Europe research and innovation program under grant agreement No. 101059548. The authors

are grateful to Peter Potapov, Co Director Global Land Analysis & Discovery group (GLAD), and the whole GLAD group for providing assistance with the Landsat ARD 16–day composites.

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
