# Peer review of "Time-series of Landsat-based bi-monthly and annual spectral indices for continental Europe for 2000–2022"

_Earth System Science Data, 2024_

## Referee Comment (RC2)

The paper proposes a new dataset of spectral indices for Europe, based on Landsat Analysis Ready Data version 2 (ARD V2) for the period 2000-2022. The dataset is fully available to the public on Zenodo, and the code is accessible on GitHub, greatly facilitating the review process. The paper is well-written.

The article's strengths are the strategies followed for building the bi-monthly composites (what authors call Tier 1) and the **extensive validation performed**. The dataset validation includes ground truth data from several sources, such as the Land Use and Cover Area Frame Survey (LUCAS), European cropland surveys (Edlinger papers), and Eurostat's tillage area statistics. These ground truth datasets are primarily used to assess the utility of the temporal-spectral feature (i.e., Tier 3, Figures 5, 6, 7, 8, and 9). Additionally, the paper evaluates annual trends in long-term temporal-spectral indices (i.e., Tier 4, Figures 10, 11, 12, 13, 14, and 15). Finally, a classification (land cover) and regression models (soil organ carbon) are trained using the dataset presented in this paper as input data, with LUCAS data as the target variable (Figures 18, 19, 20, and Tables 5, 6, 7). A statistical comparison of classes of land cover with other Landsat-derived products (i.e., EcoDataCube, Witjes et al. (2023)) was also carried out (Figure 16).

**I have some concerns about the high correlation between this paper and the Consoli et al. (2024) paper**, which is available only as a preprint and published just one month ago (https://www.researchsquare.com/article/rs-4465582/v1). However, if the **editor thinks this similarity is irrelevant**, I would **recommend a minor revision (see comments below)**.

**Major concern**

This paper, "Bi-monthly and Annual Landsat Spectral Indices for Europe 2000-2022," **is very similar to the dataset presented in Consoli et al.'s (2024) "Global Bi-monthly Landsat Aggregated Product 1997-2022. "** Both papers have not been published yet but are available as preprints.

This journal states: *"The editors encourage submissions on **original data or data collections** of sufficient quality that have potential to contribute to these aims."*

The **Consoli et al. (2024)** paper focuses on Tier 1, i.e., generating plausible aggregation and performing gap-filling in a single step by simply adjusting a convolution kernel. It introduces a new algorithm, TSIRF, which appears to outperform traditional methods like Savitzky-Golay in terms of both computation time and accuracy. The editor should consider that many authors of the **Consoli et al. (2024)** paper are also authors of this paper under review. The **Consoli et al. (2024)** paper is available as a preprint here: https://www.researchsquare.com/article/rs-4465582/v1.

The primary difference between these two papers lies in the **estimation of spectral indices**, specifically Tier 2 (Vanilla Spectral Index), Tier 3 (Temporal-Spectral Index), and Tier 4 (Long-term Temporal-Spectral Index). Essentially, **this paper builds on the data presented by Consoli et al. (2024)**. For instance, on Page 3, Line 24 **the Bimonthly aggregated cloud-optimized bands must be the same that Consoli et al. (2024) paper**.

**I believe that an ESSD brief communication, as an extension of the Consoli et al. (2024) paper—published after its acceptance—would be more appropriate for this paper, as the principal novelty is the estimation of complex temporal-spectral indices and their validation.**

**Minor comments**

- Experiments demonstrating the time-series reconstruction performance in Tier-1 would be highly relevant for readers.

- **Line 10, Page 2:** "which cover approximately 67% of the Earth's surface" – Please either remove this phrase or provide an accurate citation.

- **Line 20, Page 5:** What is SIRCLE? This term is not mentioned in either Consoli et al. (2024) or the scikit-map documentation. Please clarify the difference between SIRCLE and TSIRF.

- **Line 5, Page 6:** SIRCLE is cited as part of Consoli et al. (2024), but the acronym only appears in Figure 3 without further reference. Please ensure that SIRCLE is introduced properly if it is central to the methodology.

- **Line 20, Page 3 and Line 5, Page 36:** Referring to Witjes et al. (2023) as "limited" for presenting quarterly rather than bi-monthly maps may be a bit misleading. Especially considering the high amount of missing data at high latitudes (see Figure 1 in this paper), a more nuanced wording may be more appropriate.

- **Line 5, Page 37:** What is HLS? Please provide a brief explanation of the acronym.

- **Line 5, Page 37:** The limitations section should address the known limitations of ARD V2, such as cloud detection accuracy over Europe and the challenges in harmonizing Landsat products.

---

## Author Response (AR1)

**RC1: 'Comment on essd-2024-266', Anonymous Referee #1, 11 Oct 2024**

**Although there are many online platforms and data distribution systems for Landsat, I appreciate OpenGeoHub's effort in producing their own version of Landsat analysis-ready data. All processing code and metadata availability observe open-access principles, and this benefits the community a lot in reusing their data. The inclusion of precalculated annual and long-term indices offers an opportunity for improved environmental modeling and mapping. Also, this is a dataset that will evolve in time with new mapped years and the potential inclusion or refinement of the current list of products. That said, I don't have any major objection to its publication, but I think we would greatly benefit from further clarification, which can also improve the quality of the data and the paper.**

Thank you for recognizing the efforts of OpenGeoHub in producing a unique version of Landsat analysis-ready data. We deeply appreciate your constructive suggestions. We agree that there is room for further development, and we aim to refine and expand the data cube to make it even more robust and adaptable for various environmental monitoring and modeling applications. We hope that the data cube will prove increasingly useful to others and that it will foster additional research and applications within and beyond our current projects.

**Why did the authors produce bimonthly products rather than monthly or even 16-day intervals? One of the greatest advantages of Landsat over Sentinel and other EO data is the long time-series archive. Good data can be fetched from 1984 since Landsat-5. As this is an evolving data cube, please consider producing at least a monthly time series, as this would allow better integration with other products like climate data. This also helps in capturing temporal changes in terrestrial ecosystems.**

RE: We fully recognize the benefits of achieving monthly granularity and it is indeed our goal. Our choice of a bimonthly resolution initially came from the challenge of gap filling, as finer resolutions like monthly or 16-day intervals tend to have larger gaps due to cloud cover and other data quality issues (this issue is discussed in detail in Consoli et al. 2024). By aggregating all available scenes over a longer period, such as two months in our case, before applying gap-filling techniques, we significantly reduce the spatial gaps, thereby simplifying the gap-filling process. Additionally, for our current environmental modeling scenarios, soil organic carbon prediction and land cover classification, a two-month resolution is adequate (see e.g. Tian et al. 2024).

Our goal is to extend the temporal coverage of this data cube while achieving finer temporal resolution e.g. monthly even 16-day granularity. We are currently exploring more sophisticated gap-filling options to enable finer temporal resolutions without compromising data quality,

including utilizing spatially close valid pixels and implementing tailored gap-filling strategies for different environmental strata. We aim to continuously update and refine this data cube to support a broader range of environmental monitoring and modeling applications within and beyond OpenGeoHub's work. However, this could take months and requires significant resources; on the other hand, we believe that use of bimonthly or monthly temporal support does not affect the content of our article in the sense of methods applied, main results and data usability.

- Consoli, D., Parente, L., Simoes, R., Şahin, M., Tian, X., Witjes, M., ... & Hengl, T. (2024). A computational framework for processing time-series of Earth Observation data based on discrete convolution: global-scale historical Landsat cloud-free aggregates at 30 m spatial resolution. Submitted to PeerJ; https://dx.doi.org/10.21203/rs.3.rs-4465582/v2
- Tian, X., de Bruin, S., Simoes, R., Isik, M. S., Minarik, R., Ho, Y. F., ... & Hengl, T. (2024). Spatiotemporal prediction of soil organic carbon density for Europe (2000--2022) in 3D+ T based on Landsat-based spectral indices time-series. Submitted to PeerJ; https://doi.org/10.21203/rs.3.rs-5128244/v1

**Please clarify if the ARCO bimonthly bands represent a true measurement from a scene or a synthetic value based on statistics, like an average. If it is a synthetic value, do you expect it will reduce the capacity to assess fine spatiotemporal changes in the landscape, as the synthetic image mixes different pixels with distinct temporal and spatial indexes?**

RE: We added a clarification in the methods section (see Section 2.3.1 on Page 6). In our dataset, the ARCO bimonthly bands represent a synthetic value, specifically weighted average derived from several scenes (usually 6–7 scenes) available within and adjacent to the two-month period. The weights are assigned based on the clear sky faction of each tile, which is calculated from the number of available, non-cloudy pixels. This approach minimizes image gaps because even a single observed pixel during the period fills in the gaps; in addition use of quantiles prevents from including artifacts i.e. very high values due to uncorrected clouds or similar. This ensures that our data remains based on actual observations, not reconstructions from past data.

However, in situations where more frequent observations are crucial, this approach could potentially limit our ability to detect rapid changes. To address this concern, as suggested by the reviewer in the previous question, we are actively working to refine our methodologies and migrate to monthly composites. This will enhance temporal resolution and allow our approach to be adapted to a broader range of scenarios.

**Bare soil fraction: There are several papers indicating that NDVI alone cannot separate bare soil pixels well, as dry vegetation has a very similar spectral profile compared to soils. The classification must use a combination of indices and in some cases, land use masks. Please check the methods of Rogge et al. (2017), Diel et al. (2017), Safanelli et al. (2020), and Heiden et al. (2022).**

- *Rogge, D., Bauer, A., Zeidler, J., Mueller, A., Esch, T., & Heiden, U. (2018). Building an exposed soil composite processor (SCMaP) for mapping spatial and temporal characteristics of soils with Landsat imagery (1984–2014). In Remote Sensing of Environment (Vol. 205, pp. 1–17). Elsevier BV. https://doi.org/10.1016/j.rse.2017.11.004.*
- *Diek, S., Fornallaz, F., Schaepman, M. E., & De Jong, R. (2017). Barest Pixel Composite for Agricultural Areas Using Landsat Time Series. In Remote Sensing (Vol. 9, Issue 12, p. 1245). MDPI AG. https://doi.org/10.3390/rs9121245.*
- *Safanelli, J. L., Chabrillat, S., Ben-Dor, E., & Demattê, J. A. M. (2020). Multispectral Models from Bare Soil Composites for Mapping Topsoil Properties over Europe. In Remote Sensing (Vol. 12, Issue 9, p. 1369). MDPI AG. https://doi.org/10.3390/rs12091369.*
- *Heiden, U., d'Angelo, P., Schwind, P., Karlshöfer, P., Müller, R., Zepp, S., Wiesmeier, M., & Reinartz, P. (2022). Soil Reflectance Composites—Improved Thresholding and Performance Evaluation. In Remote Sensing (Vol. 14, Issue 18, p. 4526). MDPI AG. [https://doi.org/10.3390/rs14184526](https://doi.org/10.3390/rs14184526).*

RE: Several of the references mentioned, such as Rogge et al. (2017) and Diel et al. (2017), are already cited in our paper and have influenced our methodology. We will study and consider adding other references that you have listed above. Our use of the Bare Soil Fraction (BSF) is intended primarily as a proxy to indicate the general bareness of pixels, rather than to identify bare soil with high accuracy. Our current approach serves as an initial approach, and we plan to learn from these more specialized studies in future to more accurately quantify soil bareness. Following the suggestions and methodologies recommended by the reviewer, we have revised the corresponding content in Section 4.1 (see Page 38 Line 13-28, and Page 39 Line 4-9).

**RC2: 'Comment on essd-2024-266', Anonymous Referee #2, 07 Nov 2024**

**Major concern**

**The paper proposes a new dataset of spectral indices for Europe, based on Landsat Analysis Ready Data version 2 (ARD V2) for the period 2000-2022. The dataset is fully available to the public on Zenodo, and the code is accessible on GitHub, greatly facilitating the review process. The paper is well-written. The article's strengths are the strategies followed for building the bi-monthly composites (what authors call Tier 1) and the extensive validation performed. The dataset validation includes ground truth data from several sources, such as the Land Use and Cover Area Frame Survey (LUCAS),**

**European cropland surveys (Edlinger papers), and Eurostat's tillage area statistics. These ground truth datasets are primarily used to assess the utility of the temporal-spectral feature (i.e., Tier 3, Figures 5, 6, 7, 8, and 9). Additionally, the paper evaluates annual trends in long-term temporal-spectral indices (i.e., Tier 4, Figures 10, 11, 12, 13, 14, and 15). Finally, a classification (land cover) and regression models (soil organ carbon) are trained using the dataset presented in this paper as input data, with LUCAS data as the target variable (Figures 18, 19, 20, and Tables 5, 6, 7). A statistical comparison of classes of land cover with other Landsat-derived products (i.e., EcoDataCube, Witjes et al. (2023)) was also carried out (Figure 16). I have some concerns about the high correlation between this paper and the Consoli et al. (2024) paper, which is available only as a preprint and published just one month ago (https://www.researchsquare.com/article/rs-4465582/v1). However, if the editor thinks this similarity is irrelevant, I would recommend a minor revision (see comments below).**

**This paper, "Bi-monthly and Annual Landsat Spectral Indices for Europe 2000-2022," is very similar to the dataset presented in Consoli et al.'s (2024) "Global Bi-monthly Landsat Aggregated Product 1997-2022. " Both papers have not been published yet but are available as preprints. This journal states: "The editors encourage submissions on original data or data collections of sufficient quality that have potential to contribute to these aims." The Consoli et al. (2024) paper focuses on Tier 1, i.e., generating plausible aggregation and performing gap-filling in a single step by simply adjusting a convolution kernel. It introduces a new algorithm, TSIRF, which appears to outperform traditional methods like Savitzky-Golay in terms of both computation time and accuracy. The editor should consider that many authors of the Consoli et al. (2024) paper are also authors of this paper under review. The Consoli et al. (2024) paper is available as a preprint here: https://www.researchsquare.com/article/rs-4465582/v1. The primary difference between these two papers lies in the estimation of spectral indices, specifically Tier 2 (Vanilla Spectral Index), Tier 3 (Temporal-Spectral Index), and Tier 4 (Long-term Temporal-Spectral Index). Essentially, this paper builds on the data presented by Consoli et al. (2024). For instance, on Page 3, Line 24 the Bimonthly aggregated cloud-optimized bands must be the same that Consoli et al. (2024) paper. I believe that an ESSD brief communication, as an extension of the Consoli et al. (2024) paper—published after its acceptance—would be more appropriate for this paper, as the principal novelty is the estimation of complex temporal-spectral indices and their validation.**

RE: We acknowledge the connection between our work and Consoli et al. (2024) and the overlap in authorship due to our shared institute. However, despite this overlap, the two works provide distinct contributions, each addressing different scientific objectives.

Consoli et al. (2024) focuses on developing the TSIRF computational framework for efficient global aggregation and gap-filling of Landsat data, demonstrating this approach with a global bi-monthly aggregation of raw spectral bands (i.e., the global version of Tier 1 products referenced in our paper). In contrast, our study emphasizes Europe, with a focus on the preparation of a comprehensive data cube of biophysical indices. This includes bi-monthly time

series (Tier 2), aggregated annual series (Tier 3), and long-term temporal signatures (Tier 4). Additionally, we conduct extensive plausibility checks and machine learning applicability tests to enhance the utility and relevance of the dataset for targeted applications within Europe.

Leveraging existing datasets is a common and essential practice in advancing data processing. While this work builds on the data produced in Consoli et al. (2024), we believe this does not compromise its originality. Furthermore, all text in this manuscript, aside from standard abbreviations and input data descriptions, is original. The work of Consoli et al. (2024) has been consistently cited to ensure transparency and proper acknowledgment. We have also revised the manuscript to further clarify and highlight these distinctions (see Section 2.3.1 on Page 6).

**Experiments demonstrating the time-series reconstruction performance in Tier-1 would be highly relevant for readers.**

RE: We have incorporated performance metrics for time-series reconstruction in continental Europe, detailed in Section 3.1 and summarized in Table 4 (Page 15), with the corresponding methodology described in Section 2.4.1 (Page 10). Additional details on the testing materials and performance across various land cover classes are provided in the supplementary material to ensure the manuscript remains concise. For a broader global analysis of time-series reconstruction performance, we refer to Consoli et al. (2024). The code for this experiment has also been updated in the corresponding [GitHub repository](.).

**Line 10, Page 2: "which cover approximately 67% of the Earth's surface" – Please either remove this phrase or provide an accurate citation.**

RE: This phrase has been removed as it inaccurately quantifies the data (see Page 3 Line 16-17). In our experiments, we observed that valid pixels account for approximately 67% of the total for several years between 1997 and 2022. However, this figure does not specifically refer to cloudy pixels—it represents valid pixels, not invalid ones. Additionally, not all invalid pixels are due to cloud cover; other factors, such as atmospheric interference, snow cover and polar night in northern areas also contribute. We have removed this phrase to avoid any misunderstanding.

**Line 20, Page 5: What is SIRCLE? This term is not mentioned in either Consoli et al. (2024) or the scikit-map documentation. Please clarify the difference between SIRCLE and TSIRF.**

RE: SIRCLE was the initial name of the framework at the time of submission but has since been renamed to TSIRF. We have now updated the manuscript to reflect this change. Please refer to Section 2.2 on Page 5 for details.

**Line 5, Page 6: SIRCLE is cited as part of Consoli et al. (2024), but the acronym only appears in Figure 3 without further reference. Please ensure that SIRCLE is introduced properly if it is central to the methodology.**

RE: We sincerely thank the reviewer for spotting this. The use of the acronym SIRCLE in Figure 3 of Consoli et al. (2024) was a typographical error. This will be corrected in Consoli et al. (2024)'s final published version to align with the updated terminology.

**Line 20, Page 3 and Line 5, Page 36: Referring to Witjes et al. (2023) as "limited" for presenting quarterly rather than bi-monthly maps may be a bit misleading. Especially considering the high amount of missing data at high latitudes (see Figure 1 in this paper), a more nuanced wording may be more appropriate.**

RE: We acknowledge that referring to EcoDataCube.eu as "limited" may not fully capture its strengths and nuances. While our work provides a finer temporal resolution and additional data products, we recognize the challenges posed by high latitudes in achieving higher resolutions. We revised the wording to more accurately reflect the contributions of EcoDataCube.eu while highlighting the distinctions of our dataset. Please refer to Page 3 Line 21-23 for details.

**Line 5, Page 37: What is HLS? Please provide a brief explanation of the acronym.**

RE: We have now clarified the acronym for HLS in the revised manuscript. The updated text provides a brief explanation of HLS as the Harmonized Landsat Sentinel product, detailing its data sources and relevance. Please refer to Page 38 Line 34-35 for details.

**Line 5, Page 37: The limitations section should address the known limitations of ARD V2, such as cloud detection accuracy over Europe and the challenges in harmonizing Landsat products.**

RE: Added. The limitations section now includes the known challenges of ARD V2, such as the surface reflectance normalization used for harmonizing ARD V2, and its struggles in processing winter time images. Please refer to Section 4.4 Paragraph 1-2 on Page 41-42 for details.

**RC3: 'Comment on essd-2024-266', Anonymous Referee #3, 12 Nov 2024**

**The availability of analysable large geodata is of great importance for lowering the inhibition threshold for potential users and thus enabling informed and comprehensible decisions and political action. Against this background, the work represents an important contribution that I recommend for publication, apart from minor suggestions for changes (see below). Against the background of**

**Data-Fitness-For-Use/Data-Fitness-For-Purpose assessment approaches (Lacagnina et al., 2022; Pôças et al., 2014; Wentz & Shimizu, 2018; Yang et al., 2013) and the associated trustworthiness of geodata (products) (Lokers et al., 2016), however, I would ask the authors also to inform how metadata or additional information can help potential users to assess the suitability of the datasets for further use.**

RE: We sincerely thank the reviewer for highlighting this important aspect. We have addressed this by discussing where to find the metadata and how it can assist users in assessing the dataset's fitness for their specific applications. A brief mention is included in the introduction on Page 5 Line 1-3 to raise user awareness, and the topic is elaborated upon in the Data Availability (Section 6) on Page 44. Additionally, detailed descriptions of the metadata are provided within the public storage location of the data cube at Zenodo landing page and central access catalog. The layers will also be updated in EcoDataCube, complete with metadata and a visualization.

**Page 2, Line 28: Spelling error: Copernicus Data Spac Ecosystem ⇒Copernicus Data Space Ecosystem. Spelling error: Per-pixel count of available value ⇒Per-pixel count of available values**

RE: Both spelling errors have been corrected.

**Figure 2:  To make the manuscript easier to read, I suggest placing the figure at the beginning of shapter 2.3 and briefly explaining the basic methodological process with reference to the corresponding subsections.**

RE: We have revised the manuscript by placing Figure 1 at the beginning of Chapter 2.3 as recommended (see Figure 1 in Page 7). And we have included a brief explanation of the preparation flow with references to the corresponding subsections to enhance readability and clarity on Page 6 Line 2-7.

**Page 4, Line 20:  Could you elaborate on your perception/definition of the term (spatial) "plausibility" and differentiate it from "accuracy/uncertainty"?**

RE: In the context of our study, "plausibility" refers to the process of demonstrating the validity of the biophysical index data cube by ensuring that the data aligns logically and statistically with known land surface processes or available reference data. This term is distinct from "accuracy" or "uncertainty," as those imply direct, rigorous validation against ground-truth or survey data, which is not always feasible due to the limitations of available datasets.

As noted in the manuscript, the primary challenge lies in finding land survey data that match our data cube's temporal, spatial, and thematic coverage/focus for a true validation. Given these limitations, the "validation" in our study is performed through various methods:
 ● For BSF, NOS, and CDR, we conducted statistical analyses to assess their reliability.

- For BSF, we examined its correlation with a land survey dataset, though in a limited context—covering 2007–2016, requiring aggregation of our annual data to align with the survey, and with limited data points that do not ensure full continental EU coverage.
- For minimum NDTI, we compared it with EUROSTAT data, but only at a regional level, while our minimum NDTI layer is available at 30m resolution.

These mismatches between our high-resolution data layers and the available reference datasets make it challenging to perform a conventional "accuracy/uncertainty" analysis. Instead, we conducted a "plausibility check," which evaluates whether the data logically and statistically represent the processes they are intended to capture. This approach provides confidence in the data's usability despite the constraints of traditional validation methods. The clarification is added at the beginning of Section 2.4 Page 10 Line 9-14.

**Page 11, Line 20: Could you add a reference for "typical CRC values for each tillage type"?**

RE: The citations have been added to the manuscript on Page 12 Line 9-10. The "typical CRC values" used in our analysis were estimated based on CRC value ranges assigned according to Magdoff et al. (2000) and Zheng et al. (2022). These ranges provided a foundation for estimating the values by selecting those that maximize the correlation between WCRC and minNDTI within each NUTS2 region.

**Page 12: Could you provide a kind of principle workflow for both modelling experiments?**

RE: We have added a workflow chart (Figure 3) in the Modeling Experiments section (Section 2.6) to provide a clear overview of the processes involved in the modeling experiments on Page 13.

**Section 2.4.4: You may consider deleting section 2.4.4 or integrating elements into the results section. For example, the explanation seems somewhat contrived "These visual representations complement the statistical analysis by highlighting spatial patterns that may". In addition, the paragraph on page 12 and line 5 can be used as an introduction to the results section.**

RE: We thank the reviewer for this constructive feedback. We have made modifications to enhance the clarity and coherence of the manuscript accordingly. Specifically, we have merged key elements of original Section 2.4.4 into other relevant sections and revised the paragraph, incorporating it into the introduction of the Results section. Please see the Result section (Section 3, starting from Page 15).

**Page 13, lines 13–26: Although in my view there is no need to list the formulas of the validation metrics (F1-score, CCC), references should at least be mentioned.**

RE: References for the validation metrics have been added as suggested on Page 10 Line 20 and Page 15 Line 6.

**Page 36, lines 22–34: It is not entirely clear to me why emphasis is placed on the supposed limitations of the Bare Soil Composite (BSC). In principle, BSC represent a filtered view of the Landsat and Sentinel-2 time series with a focus on agricultural areas in order to identify stable soil patterns. The "accusation" of regional applicability also does not reflect the complexity of digital soil mapping (DSM), as the transferability of DSM approaches depends on many factors such as the representativeness of soil samples, suitable explanatory variables, or DSM models that take into account the spatial variability of soil landscapes (e.g., Broeg et al., 2024). In this respect, BSF products face the same challenge. More relevant would be a discussion of differences in the generation of BSC and BSF products. This concerns, for example, approaches to temporal-dynamic filtering taking phenology into account (Zepp et al., 2023), which would be a nice feature of your products in the future.**

RE: We sincerely thank the reviewer for their insightful comments on this matter. Our intention was not to overemphasize the limitations of the Bare Soil Composite (BSC), which we also find as highly valuable for DSM practices. Instead, our aim was to highlight the need of developing BSF products and to distinguish them from BSC. In response to the reviewer's suggestion, we have revised this section to focus on the differences in the generation and perspectives of BSC and BSF products (see from Page 38 Line 13-28).

**Page 38, lines 20-21: This result is in line with Zepp et al., 2023.**

RE: We have added the reference as suggested in Page 40 Line 33.

**Page38, section 4.3: Both use cases represent current topics. I would therefore welcome it if the discussion referred to a few relevant works.**

RE: We sincerely thank the reviewer for this valuable suggestion. We have incorporated references to several relevant works in Section 4.3 on Page 40 - Page 41.

**Page 39, lines 6ff Could you support your conclusions on the feature importance and selection together with scientific references?**

RE: In response, we have added scientific references to support our conclusions on feature importance and selection on Page 41 Line 15-23.

**References mentioned:**

- Broeg, T., Don, A., Gocht, A., Scholten, T., Taghizadeh-Mehrjardi, R., & Erasmi, S. (2024). Using local ensemble models and Landsat bare soil composites for large-scale soil organic carbon maps in cropland. Geoderma, 444, 116850. https://doi.org/10.1016/j.geoderma.2024.116850
- Lacagnina, C., David, R., Nikiforova, A., Kuusniemi, M.-E., Cappiello, C., Biehlmaier, O., Wright, L., Schubert, C., Bertino, A., Thiemann, H., & Dennis, R. (2022). Towards a data

quality framework for EOSC authorship community (tech. rep.). EOSC Association. https://doi.org/10.5281/zenodo.7515816

- Lokers, R., Knapen, R., Janssen, S., van Randen, Y., & Jansen, J. (2016). Analysis of Big Data technologies for use in agro-environmental science. Environmental Modelling & Software, 84, 494–504. https://doi.org/10.1016/j.envsoft.2016.07.017
- Pôças, I., Gonçalves, J., Marcos, B., Alonso, J., Castro, P., & Honrado, J. P. (2014). Evaluating the fitness for use of spatial data sets to promote quality in ecological assessment and monitoring. International Journal of Geographical Information Science, 28(11), 2356–2371. https://doi.org/10.1080/13658816.2014.924627
- Wentz, E. A., & Shimizu, M. (2018). Measuring spatial data fitness-for-use through multiple criteria decision making. Annals of the American Association of Geographers, 108(4),1150–1167. https://doi.org/10.1080/24694452.2017.1411246
- Yang, X., Blower, J. D., Bastin, L., Lush, V., Zabala, A., Masó, J., Cornford, D., Díaz, P., & Lumsden, J. (2013). An integrated view of data quality in Earth observation. Philosophical Transactions of the Royal Society A: Mathematical, Physical and Engineering Sciences, 371(1983), 20120072. https://doi.org/10.1098/rsta.2012.0072
- Zepp, S., Heiden, U., Bachmann, M., Möller, M., Wiesmeier, M., & VanWesemael, B. (2023). Optimized bare soil compositing for soil organic carbon prediction of topsoil croplands in Bavaria using Landsat. ISPRS Journal of Photogrammetry and Remote Sensing, 202, 287–302. https://doi.org/10.1016/j.isprsjprs.2023.06.003